evolution, genomics

rapid evolution, molecular evolution, transmissible cancer, wildlife disease, conservation genomics, RAD-capture

**Author for correspondence:**
Paul A. Hohenlohe
e-mail: hohenlohe@uidaho.edu

One contribution to a special feature 'Wild quantitative genomics: the genomic basis of fitness variation in natural populations'.

# Contemporary and historical selection in Tasmanian devils (*Sarcophilus harrisii*) support novel, polygenic response to transmissible cancer

Amanda R. Stahlke[1], Brendan Epstein[3], Soraia Barbosa[1,2], Mark J. Margres[3,4], Austin H. Patton[3,5], Sarah A. Hendricks[1], Anne Veillet[1], Alexandra K. Fraik[3], Barbara Schönfeld[6], Hamish I. McCallum[7], Rodrigo Hamede[6], Menna E. Jones[6], Andrew Storfer[3] and Paul A. Hohenlohe[1]

[1]Institute for Bioinformatics and Evolutionary Studies (IBEST), and [2]Department of Fish and Wildlife Sciences, University of Idaho, Moscow, ID 83844, USA
[3]School of Biological Sciences, Washington State University, Pullman, WA 99164, USA
[4]Department of Integrative Biology, University of South Florida, Tampa, FL 33620, USA
[5]Department of Integrative Biology and Museum of Vertebrate Zoology, University of California, Berkeley, CA 94720, USA
[6]School of Natural Sciences, University of Tasmania, Hobart, Tasmania 7005, Australia
[7]Environmental Futures Research Institute, Griffith University, Nathan, Queensland 4111, Australia

ARS, 0000-0001-5724-598X; BE, 0000-0001-7083-1588; AHP, 0000-0003-1286-9005; PAH, 0000-0002-7616-0161

Tasmanian devils (*Sarcophilus harrisii*) are evolving in response to a unique transmissible cancer, devil facial tumour disease (DFTD), first described in 1996. Persistence of wild populations and the recent emergence of a second independently evolved transmissible cancer suggest that transmissible cancers may be a recurrent feature in devils. Here, we compared signatures of selection across temporal scales to determine whether genes or gene pathways under contemporary selection (six to eight generations) have also been subject to historical selection (65–85 Myr). First, we used targeted sequencing, RAD-capture, in approximately 2500 devils in six populations to identify genomic regions subject to rapid evolution. We documented genome-wide contemporary evolution, including 186 candidate genes related to cell cycling and immune response. Then we used a molecular evolution approach to identify historical positive selection in devils compared to other marsupials and found evidence of selection in 1773 genes. However, we found limited overlap across time scales, with only 16 shared candidate genes, and no overlap in enriched functional gene sets. Our results are consistent with a novel, multi-locus evolutionary response of devils to DFTD. Our results can inform conservation by identifying high priority targets for genetic monitoring and guiding maintenance of adaptive potential in managed populations.

## 1. Introduction

Species are subject to selection by pathogens throughout their evolutionary history, shaping lineage diversification and leading to complex cellular and molecular defensive mechanisms [1]. Still, emerging infectious diseases (EIDs) can cause mass mortality and, given sufficient reproduction and genetic variation, initiate rapid adaptive evolution in a naive host population [2]. Although the prevalence and severity of EIDs in wildlife populations is now well recognized [3–6], we are just beginning to understand the evolutionary impacts of disease in wildlife. We have a relatively short recorded history of

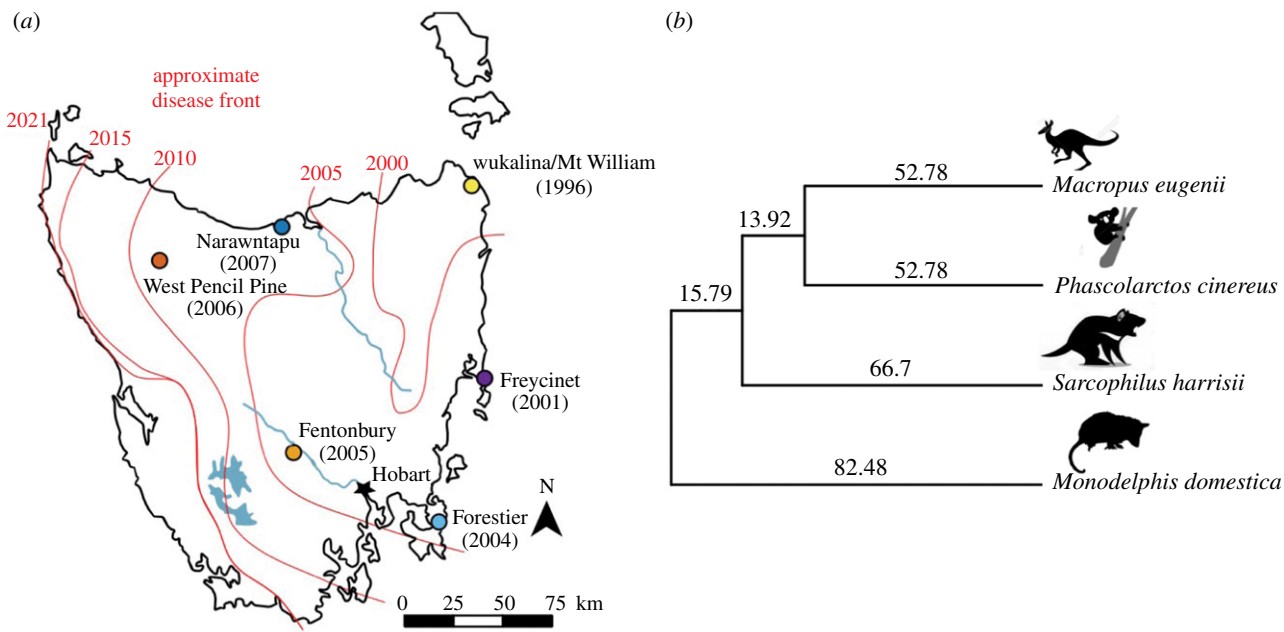

**Figure 1.** (*a*) Map of the six contemporary sampling locations relative to disease prevalance over time (red lines) with the year of first detection labled at each site. (*b*) Reduced, unrooted time-calibrated phylogeny [26] of marsupials used to estimate genome-wide historical selection on the devil lineage with estimated divergence times (Ma) indicted along edges. Devil cartoon by David Hamilton. Wallaby, koala and opossum digital images retrieved from http://www.shutterstock.com/amplicon. From top to bottom: The tammar wallaby (*Notamacropus eugenii*), koala (*Phascolarctos cinereus*), Tasmanian devil (*Sarcophilus harrisii*) and South American grey-tailed opossum (*Monodelphis domestica*). (Online version in colour.)

infectious disease in wildlife, and therefore a limited ability to predict outcomes or intervene when warranted [7,8].

High-throughput DNA-sequencing techniques and high-quality annotated reference genomes have revolutionized our ability to monitor and identify mechanisms of evolutionary responses to pathogens [8–10]. Interspecific comparisons of non-synonymous and synonymous variation (d$N$/d$S$) within protein-coding regions have long been used to identify positive selection at immune-related loci [11–13]. At the population level, rapid evolution in response to disease can be detected by tracking changes in allele frequency before, during and after the outbreak of disease [14,15]. Intra-specific comparisons across populations can reveal to what extent the evolutionary response to disease is constrained by limited genetic mechanisms or variation for adaptation [16]. Reduced representation techniques such as restriction-site associated DNA-sequencing (RADseq) [17] have made the acquisition of genome-wide, time-series genetic data more accessible in non-model systems [18]. By integrating these resources and tests of selection at differing temporal scales, we can assess whether species that show rapid evolution in response to contemporary pathogens also show evidence of historical selection to similar pathogens.

A striking example of an EID acting as an extreme selective force in wildlife is devil facial tumour disease (DFTD), a transmissible cancer first described in 1996 afflicting wild Tasmanian devils (*Sarcophilus harrisii*) [19]. Tasmanian devils are the largest extant carnivorous marsupial, with contemporary wild populations restricted to the Australian island of Tasmania. As a transmissible cancer, DFTD tumour cells are transmitted between hosts, behaving as a pathogen [20]. Transmission typically occurs as devils bite each other during the mating season after devils have reached sexual maturity [21,22]. With few notable exceptions documenting regression [23], DFTD tumours escape recognition, become malignant, and can kill their hosts within

six months [24]. Starting from a single Schwann cell origin [25], DFTD has now swept across nearly the entire species range (figure 1*a*). Devil populations have declined species-wide approximately 80% [27] with local declines in excess of 90% [28]. Nonetheless, population genomic studies have shown that devils are rapidly evolving in response to DFTD [2,29,30], and DFTD has been spontaneously cleared (i.e. regressed) in some individuals [23]. Long-term field studies and simulation modelling have predicted that cyclical coexistence or DFTD extirpation are more likely scenarios than devil extinction [31]. This is particularly alarming because devils have notoriously low genome-wide diversity, attributed to climate- and anthropogenic-induced bottlenecks [32–34]. Depleted genetic diversity at immune-related loci has probably further contributed to DFTD vulnerability [35].

Despite transmissible cancers being exceedingly rare across animals, a second independent transmissible cancer in devils, DFT2, was described in 2014 [36,37]. Comparative and functional analyses of DFTD and DFT2 showed similar drivers of cancerous mutations and tissue type of origin [38]. Low genetic diversity, chromosomal fragility [39], a reportedly high incidence of non-transmissible neoplasms [40] and injury-prone biting behaviour [41] may contribute to a predisposition to transmissible cancers in devils [42]. These findings suggest that transmissible cancers may be a recurring selective force in the Tasmanian devil lineage. If so, this leads to the hypothesis that the genes and genetic pathways associated with the ongoing evolutionary response to DFTD may have experienced recurrent historical selection in the devil lineage from previous transmissible cancers.

Because of the threat of DFTD and DFT2 to devil populations, there are ongoing conservation efforts, including the establishment of a captive devil insurance meta-population. The insurance population is managed to maintain genome-wide genetic diversity and serve as a source for re-introductions in an effort to increase genetic diversity and size of wild

**Table 1.** Number of adults sampled before and after the year of first detection of DFTD at each site. See electronic supplementary material, table S1 for sample size for each year at each locality.

| location | year of first detection | samples before | samples after | total |
|---|---|---|---|---|
| wukalina/Mt William | 1996 | 0 | 155 | 155 |
| Freycinet | 2001 | 300 | 382 | 682 |
| Forestier | 2004 | 131 | 552 | 683 |
| Fentonbury | 2005 | 99 | 169 | 268 |
| West Pencil Pine | 2006 | 52 | 348 | 400 |
| Narawntapu | 2007 | 224 | 150 | 374 |
| total | | 806 | 1756 | 2562 |

populations [43]. To inform conservation efforts, it is important to understand what types of genetic variation in natural populations may allow for evolutionary rescue from disease and maintain adaptive potential for future threats [44]. Given evidence for rapid evolution in response to DFTD, monitoring of genetic variation at candidate adaptive loci could help evaluate adaptive potential of wild populations [44,45]. In heavily managed (e.g. captive) populations, loci associated with an adaptive response to disease could be included in genotyping panels for maintaining genetic diversity [46].

Here we identify targets of selection and signatures of adaptation at both contemporary (6–8 generations) and historical (65–85 Myr) scales in Tasmanian devils. First we test for evidence of contemporary genomic response to selection by genotyping thousands of individuals sampled at several time points across six populations, using RAD-capture [47] to target nearly 16 000 loci [48]. Next we identify signatures of historical selection in the devil lineage by comparing across marsupial species with annotated genomic sequence data. Then, we test for evidence of recurrent selection by examining shared contemporary and historical signatures of selection, in terms of specific loci, genes or functional genetic pathways.

If transmissible cancer is a novel selective force acting on Tasmanian devils, we expect that genes under contemporary selection by DFTD will be different from those with a signature of historical positive selection. Alternatively, if transmissible cancer is a recurrent selective force in the devil lineage that targets the same set of genes repeatedly, we may expect a conserved response among populations and an overrepresentation of the same genes or pathways under both contemporary and historical selection. However, if there are multiple, redundant, genetic pathways that could be involved in a response to recurrent transmissible cancers, we may expect a novel response across contemporary populations and little overlap between contemporary and historical time scales. These alternatives can inform conservation efforts to manage genetic diversity for resilience in natural devil populations, and any genes or functional pathways that show both contemporary and historical selection may be relevant to cancer resistance more broadly.

## 2. Material and methods

### (a) Contemporary selection
We used the RAD-capture method (combining RADseq and sequence capture) [47] to conduct targeted genotyping of

single-nucleotide polymorphisms (SNPs) across 2562 unique individuals from multiple Tasmanian devil populations, sampled both before and after DFTD appeared in each population (figure 1a and table 1; electronic supplementary material, table S1) [30,49]. Animal use was approved under the Institutional Animal Care and Use Committee (IACUC ASAF#04392) at Washington State University and Animal Ethics Committee (A0008588, A0010296, A0011696, A0013326, A0015835) at University of Tasmania. We constructed RAD-capture libraries following Ali *et al.* [47], using the restriction enzyme *PstI* and a capture array targeting 15 898 RAD loci selected for membership in one of three functional categories: (1) those showing signatures of DFTD-related selection from previous work [2], (2) loci close to genes with known cancer or immune function, and (3) loci widely distributed across the genome (See [30,48] for more details on the devopment of this array.). See electronic supplementary material, S1 for multiplexing, read processing and SNP genotyping details.

To account for the expected high rates of genetic drift within populations, we used a composite statistic to compare signatures of selection across populations. We identified candidate SNPs as the top 1% of a de-correlated composite of multiple signals (DCMS) score [50], which combined the results of three analyses: change in allele frequency in each population after DFTD ($\Delta af$), and two methods that estimate strength of selection from allele frequencies at multiple time points in multiple populations, the method of Mathieson & McVean [14] (*mm*), which allows the estimated selection coefficient to vary over space; and *spatpg* [15], which allows the selection coefficient to vary over time and space. Individuals were assigned to generational-cohorts based on their estimated years of birth (electronic supplementary material, table S1). We estimated $\Delta af$ for five locations at which we had sampling both before and after DFTD was prevalent, according to DNA collection date and estimated date of birth, combining multiple cohorts when applicable (table 1; electronic supplementary material, table S1). Both time-series methods (*mm* and *spatpg*) incorporate estimates of effective population size, which ranged from 26 to 37 according to a two-sample temporal method [51,52] (electronic supplementary material, table S3). DCMS reduces the signal-to-noise ratio by combining *p*-values from different tests at each SNP while accounting for genome-wide correlation among statistics. We included SNPs with results from at least 11 of the 12 individual tests ($\Delta af$ for five populations, *mm* for all six populations and *spatpg*) and weighted based on the statistics with results at that SNP. To characterize the role of standing genetic variation in rapid evolution [53], we visualized the initial allele frequencies of each population for each analysis of contemporary evolution (electronic supplementary material, figures S2 and S9). We evaluated repeatability among populations by comparing population-specific *p*-values of $\Delta af$ and *mm* with the R

package dgconstraint for a similarity index called the C-score, where 0 indicates no similarity between populations [16]. See electronic supplementary material, S1 for details of each analysis.

## (b) Historical selection

We combined existing genomic resources for the South American grey-tailed opossum (*Monodelphis domestica*) [54] and tammar wallaby (*Notamacropus eugenii*) [55] from the Ensembl database [56] and the recently published transcriptome assembly of the koala (*Phascolarctos cinereus*) [57] to identify genome-wide signatures of positive selection in devils, relative to these other species using the branch-site test of PAML (phylogenetic analysis by maximum likelihood) [58,59]. We compiled alignments of orthologous genes and reduced the marsupial time-calibrated phylogeny of Mitchell *et al.* [26] to those species for which annotated full genomes are available (figure 1*b*). The branch-site test compares likelihood scores for two models which estimate $dN/dS$ among site classes of a multi-sequence alignment, allowing $dN/dS$ to exceed 1 (positive selection) in a proportion of sites along a single branch in the alternative model. We reduced the potential for false positives by filtering any putative orthologues with extreme sequence divergence ($S > 2$), measured as the sum of synonymous mutations per gene ($S$), and ensuring alignments of nucleotides were longer than 100 bp [60,61]. We identified historical candidates with the likelihood-ratio test, comparing the likelihoods of the alternative and neutral models with 1 d.f. and $\alpha = 0.05$. Historical candidates were those with estimates of $dN/dS > 1$ along the devil branch and FDR > 0.05 after correcting for multiple testing [62]. See electronic supplementary material, S1 for details regarding orthology identification and PAML implementation.

## (c) Recurrent selection

We refer to genes under both contemporary and historical selection as candidates for recurrent selection. To test whether genes under contemporary selection differed from genes under historical selection, we first tested for significant overlap with Fisher's one-tailed test. To test for differences in the strength of selection, we compared the distributions of $dN/dS$ and the proportion of sites per gene found under positive selection among candidates for recurrent selection to all other historical candidates from the genome-wide background using non-parametric tests of equality, the Kolmogorov–Smirnov test [63], which is more sensitive to the centre of the distributions, and the Anderson–Darling test [64], which is more sensitive to extreme values of the distribution and often has more power. To identify and compare key mechanisms of adaptation among candidate genes from each set, we used gene ontology (GO) term enrichment analysis using the SNP2GO package [65], the PANTHER web-interface [66], and in gene sets of the molecular signatures database (MsigDB), using the subset of genes tested for each test as the respective background set [67]. We capitalized on the wealth of ongoing research in devils and DFTD by comparing our contemporary and historical candidates to those previously identified using different datasets and analytical approaches [2,29,30,48,68]. See electronic supplementary material, S1 for details of these comparisons.

# 3. Results

## (a) Genomic data

To test for contemporary selection, we sampled a total of 2562 individuals across six localities of Tasmania before and after DFTD prevalence (table 1 and figure 1*a*; electronic supplementary material, table S1), with a RAD-capture array [48]. After filtering, we mapped a total of 517.7 million reads against targeted loci. The mean final coverage of targeted loci was 14.8×, with 76.6% of all samples having coverage of at least 5× (electronic supplementary material, figure S1). After filtering, we retained 14 585–22 878 SNPs for downstream analysis, depending on the sampled time point and population.

## (b) Evidence for contemporary selection

Among each elementary test for selection signatures, 161–232 SNPs (depending on population) were in the top 1% of allele shifts following disease ($\Delta af$), 209–217 were in the top 1% of $mm$ scores, and 213 were in the top 1% of $spatpg$ scores (electronic supplementary material, table S4, figures S7–S8). Across populations and elementary tests for contemporary selection ($\Delta af$, $mm$, $spatpg$), $p$-values were not correlated (Pearson's $r < 0.155$ for all tests; electronic supplementary material, figure S10). The computed repeatability indexes for population-specific responses $\Delta af$ and $mm$ were $C_{\Delta af} = 4.86$ ($p = 1 \times 10^{-4}$) and $C_{mm} = 3.72$ ($p = 1 \times 10^{-4}$), which implies a low, but significant level of repeatability [16]. In the top 1% of DCMS scores (greater than or equal to 1.167), we identified 144 candidate SNPs for contemporary selection by DFTD; of these, 79 had annotated genes (186 total) within 100 kb (figure 2; electronic supplementary material, table S5). The initial frequencies for candidate SNPs were not consistently skewed toward intermediate frequencies across all populations (electronic supplementary material, figure S9). The skew we observed in a few populations (e.g. Fentonbury, Narawntapu) may reflect differences in the power to detect selection with low versus intermediate minor allele frequencies within those populations, or possibly other factors such as balancing selection [53].

Comparing our contemporary candidates and those previously identified in devils with selection and genome-wide association analyses [29,30,48,68], we found many overlapping genes (discussed below). Notably, we found significant enrichment of candidates previously associated with DFTD-related phenotypes in females (14 genes, $p = 4.2 \times 10^{-8}$, odds ratio = 7.3) [48]. GO enrichment analysis found middle ear morphogenesis (GO:0042474) significantly enriched among contemporary candidate SNPs (FDR < 0.05). Five candidate SNPs were within the 100 kb window of two genes associated with this term: EYA1 and PRKRA. Both EYA1 and PRKRA are involved in cell proliferation and migration and implicated in tumour suppression and angiogenesis [69–71].

## (c) Evidence for historical selection

Of the 18 788 genes annotated in the devil reference genome, 6193 had 1-to-1 orthologues in at least three of the four marsupial genomes and an appropriate sequence divergence ($S < 2$). Using the branch-site test for positive selection in PAML, we found a total of 1773 genes to be candidates for historical positive selection (electronic supplementary material, table S6). Estimates of $dN/dS$ spanned the full range of possible values, from 1.05 to 999 and proportion of sites with substitutions per gene ranged from 0.01 to 0.78 (figure 3). The majority of genes were classified as having a molecular function of binding (GO:0005488) or catalytic activity (GO:0003824); a plurality involved in cellular

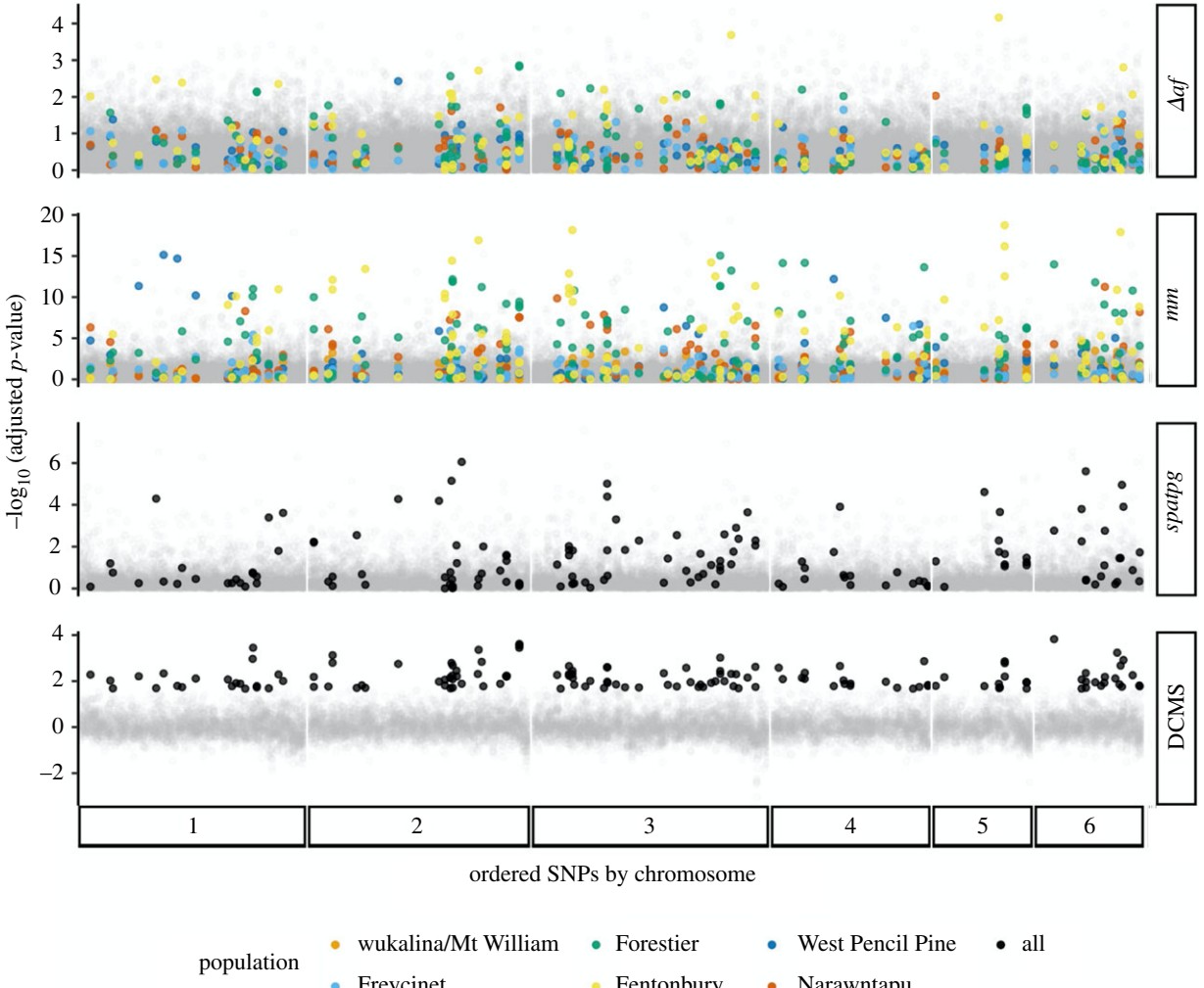

**Figure 2.** Results of each elementary test of contemporary selection across populations and the composite scores for final candidate (filled points) and non-candidate (opaque grey points) SNPs, ordered by chromosome and coloured by population when applicable. From top to bottom: Change in allele frequency ($\Delta af$), Mathieson & McVean (*mm*) [14], *spatpg* [15], *DCMS* [50]. (Online version in colour.)

processes (GO:0009987) or biological regulation (GO:0065007); and a plurality as participating in the Wnt signalling pathway (P00057). None of these pathway classifications were significantly enriched.

### (d) Recurrent selection

Of the 186 contemporary candidate genes, 68 had 1-to-1 orthologues among the four marsupials and were tested for historical selection. Sixteen genes showed evidence of historical selection and are thus candidates for recurrent selection ($dN/dS > 1$, FDR < 0.05; electronic supplementary material, table S6). Contemporary candidates were not enriched for historical selection according to Fisher's test (Odds ratio = 0.0, $p = 1$). Among the 16 recurrent candidates, $dN/dS$ estimates spanned 15.7–999 and proportion of sites per gene 0.01–0.25 (figure 3). According to the Anderson–Darling and Kolmogorov–Smirnov tests of equality, neither distributions of $dN/dS$ estimates (figure 3a; A.D. $p = 0.86$; K.S. $p = 0.58$), nor proportion of sites (figure 3b; A.D. $p = 0.49$; K.S. $p = 0.48$) differed between candidates for recurrent selection (in black) and historical candidates (in red).

After correcting gene set enrichment for multiple testing (FDR < 0.05) and requiring at least 10 genes in the background set, we did not find functional enrichment of any MSigDB gene sets among recurrent candidates or shared

between both contemporary and historical sets. Importantly, the permutation test of shared gene sets found *fewer* shared between historical and contemporary selection than expected by chance ($p < 0.001$; electronic supplementary material, figure S11).

## 4. Discussion

### (a) Contemporary responses to DFTD

Using a targeted set of nearly 16 000 loci, we detected widespread evidence of a response to selection by DFTD across the Tasmanian devil genome. Our results extend previous work that has shown genomic evidence of a response to DFTD in wild populations [2,29,30,72]. Here we greatly increased the sample size of individuals and genetically independent populations for greater power, resulting in strong evidence of a response to selection widely distributed across the genome. We found greater similarity across populations within analytical approaches than among methods within populations and relatively low, but significant repeatability across populations. This result is consistent with rapid, polygenic evolution facilitated by selection for standing variation within populations that was present prior to disease arrival. This time scale (three to eight generations) would

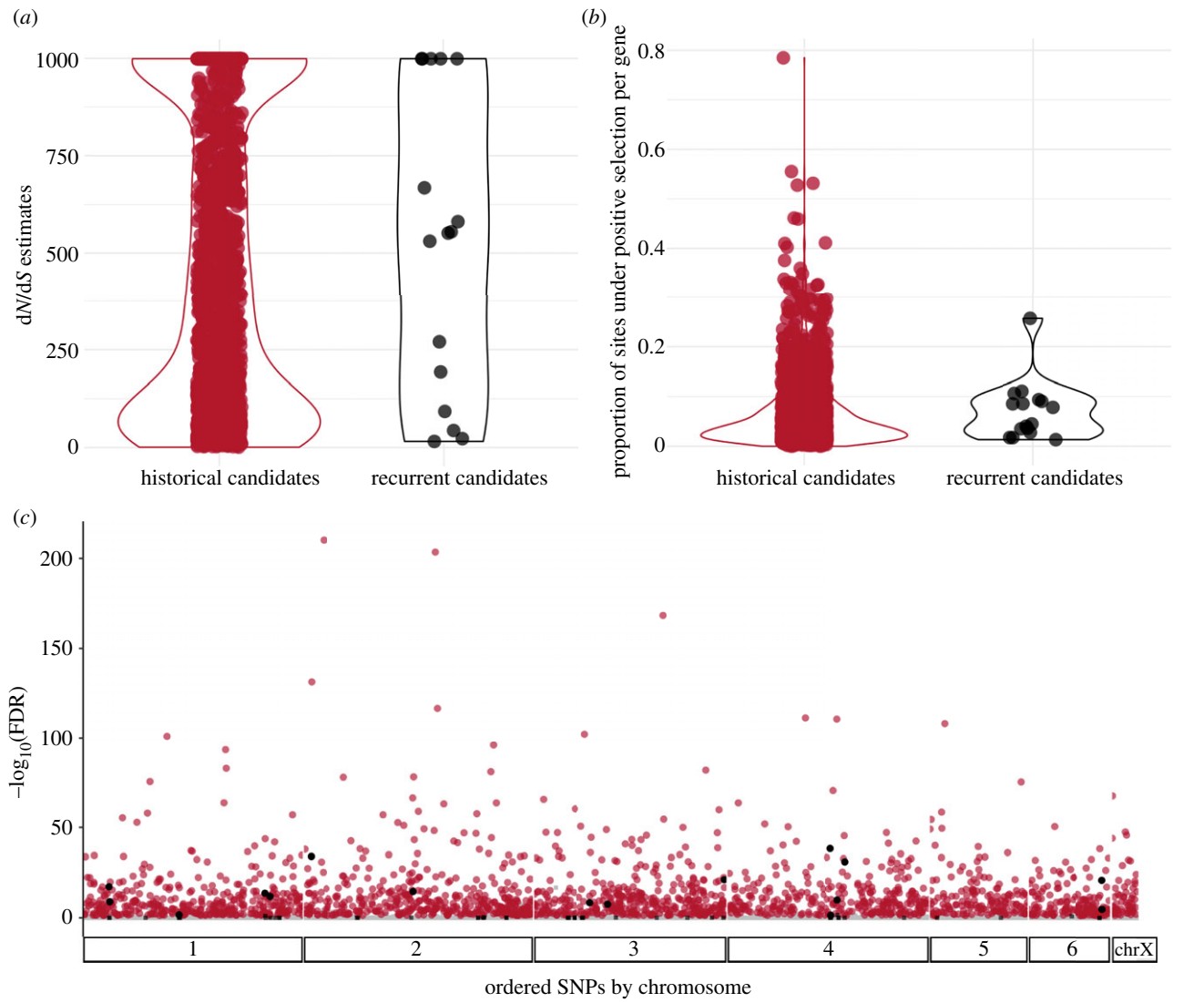

**Figure 3.** (a) Estimates of dN/dS and (b) the proportion of sites under positive selection [58] for historical candidates across the genome-wide background (red; N = 1982) and candidates for recurrent selection (genes with significant results for both historical and contemporary selection; black). Each point represents the respective result for a single gene. (c) Distribution of −log(FDR) for historical selection across all 6193 genes tested (grey squares, non-significant at both scales; black squares, significant contemporary and non-significant historical; red circles, non-significant contemporary and significant historical; black circles, significant at both scales). (Online version in colour.)

likely be too short for new mutations or migration to play a substantial role in DFTD response, and genetic variation is shared across the species range, despite geographical population structure [30,73].

In line with previous population genomic studies [2,29,30,48,68], our analysis of contemporary evolution detected a putatively adaptive response related to the immune system, cell adhesion and cell-cycle regulation (electronic supplementary material, table S5). Our GO enrichment result for middle ear development (GO:0042474) among contemporary candidates may highlight selection for interactions with the peripheral nervous system and cell proliferation. Genes annotated with nervous system associations may indicate selection for behavioural changes [29], or highlight importance and vulnerability of peripheral nerve repair by Schwann cells in devils, given the prevalence of biting and Schwann cell origin of DFT [42]. Significant overlap for genes associated with devil infection status (case-control), age, and survival [48] among our contemporary candidates is a strong indicator that these contemporary candidates likely confer relevant phenotypic change. We

also confirmed five (CRBN, ENSSHAG00000007088, THY1, USP2, C1QTNF5) of seven candidates identified previously [2] in a genome scan for loci under selection from DFTD in three of the same populations (Freycinet, Narawntapu and West Pencil Pine). By contrast, we identified those five and only two more (TRNT1 and FSHB) of 148 candidates from a re-analysis of that same dataset which studied population-specific responses [29].

Among genes that have been associated with host variation responsible for tumour regression on devils [68,74], we found only JAKMIP3, a Janus kinase and microtubule binding protein [74], in our list of contemporary candidates. However, we found devil regression candidates TL11, NGFR and PAX3, which encodes a transcription factor associated with angiogenesis [75]; as well as GAD2, MYO3A and unannotated ENSSHAG00000009195 [74], among population-specific candidates for allele frequency change (Δaf), possibly reflecting differences in test sensitivities. Overall, the paucity of candidates shared between our contemporary analysis and regression studies suggests that regression may not be the dominant form of phenotypic response to DFTD;

to date tumour regression has only been detected in a few populations [74] not represented in our study.

## (b) Historical selection in the devil lineage

With our genome-wide molecular evolution approach [58], we found widespread historical positive selection across the devil genome in about 28% of all 6249 orthologues tested (electronic supplementary material, table S6). The branch-site test is known to be less conservative than related models, particularly when divergence among species is large [76], but the rates of historical selection we found in devils are similar to those described in other taxa (e.g. 23% of genome-wide orthologues among 39 avian species using a similar approach) [13].

We did not find preferential positive selection for immunity-related genes, as has been shown in primates [1], eutherian mammals more generally [77] and birds [13]. Instead, we found the highest proportion of pathways under historical selection to be functionally classified within the Wnt pathway, a signalling cascade regulating cell adhesion and implicated in carcinogenesis [78]. As genomic resources grow and improve in marsupials [10], interspecific analyses for positive selection at finer scales may reveal more recent and specific selection targets in Tasmanian devils. Our ability to detect historical selection due to transmissible cancer in devils could be improved by genome assembly efforts among more closely related Dasyuridae, as well as complementary annotation.

## (c) Comparing contemporary and historical time scales

Remarkably few transmissible cancers have been discovered in nature [79,80], and yet two of those independent clonally transmitted cancers have been discovered in Tasmanian devils in less than 20 years. This and the observed rapid evolutionary response to disease suggest that transmissible cancers may be a recurrent event in devils. We found no significant overlap of historical and contemporary selection at either individual genes or functional gene sets. This does not rule out the possibility of prior transmissible cancers in devils; but it suggests that if transmissible cancers have been a recurrent feature of devil evolution prior to DFTD, they did not generally impose selection on the same set of genes or genetic pathways that show a contemporary response to DFTD. Nonetheless, the 16 candidate genes showing both historical and contemporary evidence for selection (electronic supplementary material, table S7) raise interesting targets for understanding adaptively important variation in devils.

The 16 candidate genes for recurrent selection (electronic supplementary material, table S7) are generally related to three main themes: transcription regulation, the nervous system and the centrosome. Four of these candidates for recurrent selection were previously associated with disease-related phenotypes [48]. We additionally found 82 historical candidates previously identified in the top 1% of SNPs associated with disease-related phenotypes with three represented in the top 0.1% associated with large effect sizes for female case-control and survival [48]. This overlap lends support to the hypothesis of recurrent selection by transmissible cancers, but was not significant ($p = 1$, odds ratio = 0). Both our contemporary selection analysis and the genome-wide association study (GWAS) approach used by

Margres and colleagues [48] are statistically limited by small populations, sample size and the time scale over which DFTD-related selection has occurred. By considering the complement of these results together, the overlapping historical, GWAS, and contemporary candidates may still be targets of recurrent selection along similar functional axes, potentially including transmissible cancer.

The low prevalence of candidates for recurrent selection and lack of shared functional gene set enrichment between both contemporary and historical signatures of selection suggest a novel response to DFTD compared to historic selection in the devil lineage. However, there are alternative hypotheses. For example, there could be redundancy in genetic mechanisms underlying resistance to transmissible cancers, potentially as a result of repeated selection for resistance, allowing selection to act across many loci [81]. That is, the low genetic diversity observed in devils could be the result of widespread historical purifying selection resulting from transmissible cancers or other diseases [82], or historical bottlenecks due to climate change and habitat loss [32–34], that prevent a response to selection under DFTD at loci that are still associated with disease phenotypes.

The widespread contemporary evolution we found in devils reflects the recent prediction [83] that response to an emergent disease is most likely to be controlled by many genes conferring quantitative resistance [84], for example, by reducing the within-host growth rate of tumours. DFTD is predicted to become less virulent in the short-term [31,85]. If DFTD persists long-term in the devil population with ongoing coevolution, it may lead to diversifying selection for specific, qualitative host resistance mechanisms [83]. Indeed, phylodynamic analysis of DFTD as it spread across Tasmania supports the hypothesis that devils may be mounting a response; transmission rates have decayed such that DFTD appears to be shifting from emergence to endemism [85]. Although host-genomic variation was not jointly considered in that study, the combined evidence of multiple studies demonstrating rapid evolutionary response of devils to DFTD, including this one, support these interpretations.

## (d) Conservation implications

Calls have been made to consider the historical context of adaptation when proposing conservation management solutions based on genomic results [86]. Our analysis of historical selection largely supports the hypothesis that DFTD is a newly emerging and novel selective force, distinctly shaping today's remaining wild devils. The targets of novel selection that we identified (figure 2; electronic supplementary material, table S4) and their functional roles should be considered for prioritization of monitoring and conservation in light of DFTD. At the same time, the wide distribution of contemporary candidates across the genome also highlights the importance of standing genetic variation to continue to respond to unique selective forces, including local environmental factors [30]. Genomic monitoring could be useful for maintaining both functional diversity at candidate loci and genome-wide variation in captive populations [46,87,88] and in the wild. Multiple genomic tools are available for targeted monitoring of large sets of loci (e.g. [89,90]) and could be used to track adaptive evolution and potential in the form of genetic diversity [44]. However, before management decisions are made for specific genes,

further work would need to identify favoured alleles and fitness effects for the genes we identified (electronic supplementary material, table S5).

DFTD has yet to reach devils in the far west (figure 1*a*) and continues to circulate throughout the island. To maintain long-term adaptive capacity in the face of similar recurrent selective forces including DFT2 and potential future transmissible cancers, our results warrant (1) the monitoring of genetic variation in broad functional groups and (2) management strategies to maintain genetic diversity across those broad groups. Although these populations were not subject to DFT2 at the time of writing, an important and interesting future direction should examine the evolutionary response to DFT2 and could compare loci under selection by the two independent transmissible cancers. This study could provide a list of candidate loci for development of a genotyping panel for either purpose, with flexibility to target many or fewer loci. At the same time, given urgent and unpredictable present-day threats including not just emerging diseases but environmental change and population fragmentation, it is important that monitoring and population management also focus on maintaining genetic variation across the genome.

# 5. Conclusion

Our results suggest that the contemporary evolutionary response to DFTD is mostly novel compared to the genome-wide signature of historical selection. Comparing the degree of overlap and distributions among contemporary and historical candidates did not support recurrent selection on a common set of genes in response to transmissible cancer. Our work contributes to mounting evidence of possible mechanisms by which devil populations are persisting and rapidly evolving in the face of DFTD despite overall low genetic diversity and population bottlenecks [2,23,48,72,91]. Broadly, this type of approach can be applied to analyses of novel threats in wildlife populations in the current era of anthropogenic global change to guide monitoring and management actions focused on genetic adaptive potential.

Ethics. Animal use was approved under the Institutional Animal Care and Use Committee (IACUC ASAF#04392) at Washington State University and Animal Ethics Committee (A0008588, A0010296, A0011696, A0013326, A0015835) at University of Tasmania.

Data accessibility. Demultiplexed sequence data has been deposited at NCBI under Bio-Project PRJNA306495 (http://www.ncbi.nlm.nih.gov/bioproject/?term=PRJNA306495) and BioProject PRJNA634071 (http://www.ncbi.nlm.nih.gov/bioproject/?term=PRJNA634071). Code and tabular results are available at https://github.com/Astahlke/contemporary_historical_sel_devils and on the Dryad Digital Repository: https://doi.org/10.5061/dryad.jq2bvq872 [92].

The data are provided in electronic supplementary material [93].

Authors' contributions. A.R.S.: conceptualization, data curation, formal analysis, investigation, methodology, resources, visualization, writing-original draft, writing-review and editing; B.E., S.B. and M.J.M.: data curation, formal analysis, methodology, writing-review and editing; A.H.P.: methodology, writing-review and editing; S.A.H.: conceptualization, methodology, writing-review and editing; A.V.: investigation, writing-review and editing; A.K.F.: data curation, methodology, writing-review and editing; B.S.: investigation, resources; H.I.M.: conceptualization, funding acquisition, project administration, writing-review and editing; R.H.: funding acquisition, investigation, project administration, resources, writing-review and editing; M.E.J.: conceptualization, funding acquisition, investigation, methodology, project administration, resources, writing-review and editing; A.S.: conceptualization, funding acquisition, investigation, methodology, project administration, supervision, writing-review and editing; P.A.H.: conceptualization, data curation, formal analysis, funding acquisition, investigation, methodology, project administration, resources, supervision, writing-review and editing.

All authors gave final approval for publication and agreed to be held accountable for the work performed therein.

Competing interests. We declare we have no competing interests.

Funding. This work was funded by NSF grant no. DEB-1316549 and NIH grant no. R01-GM126563 to A.S., P.A.H., M.E.J. and H.I.M. as part of the joint NSF-NIH-USDA Ecology and Evolution of Infectious Diseases program. Genomics and bioinformatics were supported by an Institutional Development Award (IDeA) from the National Institute of General Medical Sciences of the NIH under grant no. P30 GM103324. This work was supported by the Bioinformatics and Computational Biology Program at the University of Idaho in partnership with IBEST (the Institute for Bioinformatics and Evolutionary Studies). Sample collection was funded under Australian Research Council grant nos. DE170101116, DP110102656, LP0989613 and LP0561120 to R.H., M.E.J., H.I.M. and A.S., an ARC Future Fellowship FT100100031 to M.E.J., and multiple awards from the University of Tasmania Foundation Eric Guiler Research Grant.

Acknowledgements. The authors would like to acknowledge Michael R. Miller, Sean M. O'Rourke and Cody G. Wiench for their contributions to the RAD-capture data; and Rebecca Johnson and Denis O'Meally for providing the koala transcriptome data.

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
