## [Peer Review File · Proceedings of the Royal Society B: Biological Sciences]

Review History

RSPB-2020-1851.R0 (Original submission)

Review form: Reviewer 1

Recommendation

Major revision is needed (please make suggestions in comments)

Scientific importance: Is the manuscript an original and important contribution to its field?

Good

General interest: Is the paper of sufficient general interest?

Good

Quality of the paper: Is the overall quality of the paper suitable?

Acceptable

Is the length of the paper justified?

Yes

Should the paper be seen by a specialist statistical reviewer?

No

Do you have any concerns about statistical analyses in this paper? If so, please specify them explicitly in your report.

No

It is a condition of publication that authors make their supporting data, code and materials available - either as supplementary material or hosted in an external repository. Please rate, if applicable, the supporting data on the following criteria.

Is it accessible?

Yes

Is it clear?

Yes

Is it adequate?

Yes

Do you have any ethical concerns with this paper?

No

Comments to the Author

My comments to the authors are attached as a pdf file.

Review form: Reviewer 2

Recommendation

Major revision is needed (please make suggestions in comments)

Scientific importance: Is the manuscript an original and important contribution to its field?

Good

General interest: Is the paper of sufficient general interest?

Excellent

Quality of the paper: Is the overall quality of the paper suitable?

Acceptable

Is the length of the paper justified?

Yes

Should the paper be seen by a specialist statistical reviewer?

No

Do you have any concerns about statistical analyses in this paper? If so, please specify them explicitly in your report.

Yes

It is a condition of publication that authors make their supporting data, code and materials available - either as supplementary material or hosted in an external repository. Please rate, if applicable, the supporting data on the following criteria.

Is it accessible?

Yes

Is it clear?

Yes

Is it adequate?

Yes

Do you have any ethical concerns with this paper?

No

Comments to the Author

The authors synthesize results from a number of population genomic approaches aimed at detecting selection at different timescales in the Tasmanian Devil, which is facing a presumably novel selective pressure due to transmissible cancer across much of its range. I found the study system and design of the experiment to be very interesting, but I don't think it is suitable for publication in its current form. Particularly, I found the methods and results section to be critically lacking in detail.

Prior to going to the supplemental, I found it hard to understand what was being compared with what and how comparisons were being made. I recognize that space constraints exist, but in this case, I can't even tell from the Methods what criteria are being used to determine candidate SNPs. I don't think a reader should be expected to pull up the Supp. Materials to get this kind of basic information. In terms of raw number of lines, over half of the manuscript is dedicated to the Discussion, so if space is a concern, I think space can be carved out for additional detail by slimming down the Discussion.

The Results are additionally peppered with very vague statements like "The distributions...had an overall low magnitude of dN/dS and a low proportion of sites." Given that these are metrics with meaning for most population genomics people, why not provide the actual numbers? This lack of detail is further reflected in the figures. Figure 3 provides very little information regarding both the absolute and relative magnitudes of the distributions being compared. There is lots of discussion regarding specific genes, pathways, GO categories, etc., but none of this is tied to any of the information in Figure 2.

The main result of the paper seems to be lower than expected overlap between candidate loci identified via historical and contemporary selection analyses. However, given that candidates are defined somewhat arbitrarily, it makes me wonder how robust this result would be to changes in the criteria used for designating candidates. Also, while much of the results/discussion focus on comparing the current results to prior studies, I often found it difficult at times to clearly identify the novel findings of this study vs what was found in older studies. With regard to contemporary selection, the authors mention in multiple places that this study greatly expands the number of individuals and genetically independent populations from prior studies, but I couldn't get a sense for what samples and populations were unique in the current study, nor could I get a clear sense of how this additional data has allowed for any new insight. Lastly, the mention of genetically independent populations had me hoping for more insight into the extent that the selection response was shared (or not) across populations. After reading the supplemental materials, it seems this may be a limitation of the method, but I was left wondering throughout the manuscript when the authors would address this with the data. Overall, I think there is a lot of good content in the paper, and with additional detail and edits could be made suitable for publication. There are some angles that I think would benefit the would like to see fleshed out, particularly inter-population variability or consistency, but I understand if this is outside of the authors' intended scope and editors' time constraints for the special feature.

Specific comments:

Line 109: Might be nice to give ballpark indication of what is meant by contemporary and historical in this study.

Line 112: RAD-capture panel was introduced in prior paragraph. Seems unnecessary here.

Line 113: shared between what? I couldn't tell if this meant looking for sharing of these signals across populations or if it meant sharing across the timescales.

Line 115: I got hung up here with how the hypotheses are set up and/or contrasted. I think part of what hung me up is that it isn't clear if we are talking about transmissible cancer *sensu stricto* DFTD or more broadly. It would seem to me that the novelty of the selective force as it pertains to the selection response would come primarily from the specific type of cancer, and there are many ways that even a particular cell lineage can become malignant.

Does non-overlap of selection targets between historical and contemporary selection necessarily imply novelty of transmissible cancer? Cancer related pathways can be so diverse, it wouldn't surprise me if recurrences of transmissible cancer invoked selection responses at different sets of loci. This seems covered by third hypothesis, but doesn't this just mean that non-overlap does not directly connect to novelty vs recurrence?

I guess I don't see the need to tie the interpretation (novelty vs recurrence) into the basic hypothesis being tested: overlap vs non-overlap of genes involved in historic vs recurrent selection. The latter results are interesting in and of themselves, and they certainly have some bearing on novelty, but I find linking them together like this to be somewhat distracting and confusing.

Line 140-148: Would be good to have a bit more of an indication as to the actual criteria that was used to be considered a candidate. It says that the 3 tests were combined, but I can't tell how this was used to determine if gene is a candidate.

Line 142: Why not delta-p?

Line 164: I stumbled on the first sentence, which I think caused me to struggle on understanding the subsequent content. I initially thought the authors were testing whether the "number", as in the count, of selected genes differs from genome-wide background, and I could not understand what genome-wide background could mean in this case. Upon reading a second time, it seems "number" is being used in the sense of 'the collection of' genes under contemporary selection. Consider rewording.

Line 166: Differences between what? After reading full MS, I understand this better now. I think my confusion was persisting from the preceding sentence, where I thought the authors were testing for a difference in the actual number of genes.

Line 183: What percentage of raw SNPs is this?

Line 187: State 1% in methods. I sympathize that these percentile cutoffs are inherently somewhat arbitrary, but how sensitive are results to the cutoff used?

Line 189: How is enrichment being done? Are GO terms from the devil annotation? Are Rapture probes biased toward disease-related genes? Does background gene set take into account these biases? I ended up finding this info in Supp Mat, but might be nice to have a nod to this in main doc.

Line 200: Again, what was criteria for inclusion? This really ought to be in the main text. Also, why was no multiple test correction done for the historical selection analysis? Is this not typical for this type of analysis? Seems like it would be called for in this case.

Line 211-212: What does overall low magnitude mean? I can't glean from text or figure anything regarding the magnitudes of candidates from genome-wide.

Line 212-214. Provide numbers even if a negative result.

Line 222: How did you correct for multiple testing?

Line 237: Wondering whether sensitivity of historical selection tests is sufficient to claim that a lack of signal justifies the positive assertion that selection has NOT occurred. Seems like more appropriate to say that the contemporary candidates do not show evidence of historical selection.

Line 266: What is meant by genes associated with devil case-control?

Line 270: What populations? Is this a sample size/power issue since this differed between populations?

Line 280: Is PAX3 a devil tumour regression candidate or just a cancer-related gene?

Line 304: Analytical resolution?

Line 311: What is the number? I'm not sure what a "small handful" of bivalve molluscs would be.
Line 349: See comment regarding line 222.

Supplemental:

What is Bonferroni FDR?

I'm confused by the inflation factors. Where are these Z-scores coming from? And shouldn't they be squared when calculating a GIF?

There are a number of typos in the Supplemental text and figure/table captions.

Supp. Table S7: Would be nice to have the p-values from the historical selection analysis for each entry in the table.

Line 459: Reads funny to me.

Figure1B: Would be nice to have an estimate of divergence times at the nodes.

Review form: Reviewer 3

Recommendation

Major revision is needed (please make suggestions in comments)

Scientific importance: Is the manuscript an original and important contribution to its field?

Good

General interest: Is the paper of sufficient general interest?

Excellent

Quality of the paper: Is the overall quality of the paper suitable?

Good

Is the length of the paper justified?

Yes

Should the paper be seen by a specialist statistical reviewer?

No

Do you have any concerns about statistical analyses in this paper? If so, please specify them explicitly in your report.

Yes

It is a condition of publication that authors make their supporting data, code and materials available - either as supplementary material or hosted in an external repository. Please rate, if applicable, the supporting data on the following criteria.

Is it accessible?

Yes

Is it clear?

Yes

Is it adequate?

Yes

Do you have any ethical concerns with this paper?

No

Comments to the Author

In this study, Stahlke et al. study the rapid evolution of a transmissible cancer in Tasmanian devils. This transmissible cancer has been evolved independently twice (DFTD and DFT2), which the authors use to see whether the same loci were under selection (as a response to the transmissible cancer) in Tasmanian devils. The authors use two approaches: they look for contemporary selection using allele frequency changes over a shorter timescale, as well as looking for signatures of selection using a molecular evolution approach. The authors compare the overlap in the targets of selection, and while they do find some genes under selection across both tests of selection, they find that there was less overlap than expected by chance. Overall, I found this to be a nice study on an important and interesting topic for both evolution and conservation. However, there were a few methodological issues the author should address before I can accept it.

Major Comments:

- One main concern is that their Δaf method is not using Fisher's angular (or arcsine) transformation. The magnitude of allele frequency change is strongly mediated by where the frequency is in the frequency spectrum; allele frequency changes are larger when they are more intermediate. Fisher and Ford (1947) is the original reference, but see Kelly and Hughes (2019, Genetics) for a recent clear application for the arcsine transform. I like the DCMS approach the authors use, and I imagine this sort of averaging across approaches means their findings are robust despite not using the arcsine transform. Still, using raw allele frequency change is not optimal and should be corrected, especially considering they ranked everything to get a pseudo-p-value for their DCMS input. As an aside, it would be interesting to know if their present ranked SNPs are enriched for more intermediate frequencies due to not using the angular transform.
- Lines 141-143: the authors say that they measured Δaf in each population. First, the authors need to specify how these data within each population are combined together to form the pseudo-p-value, in either the main text or Supplementary Materials. Second, this allows for a possibly interesting alternative analysis: why not look for convergence across populations in allele frequency change? Do certain SNPs show convergence in allele frequency change direction, across populations? For example, see the idea of convergence correlations in Buffalo and Coop, 2020, PNAS – this method isn't needed here, but the same idea and signal is worth looking at. Note though that gene flow makes this difficult, which relates to my next point.
- One concern I had is gene flow across the five locations with sampling before/after DTFD. This can also lead to significant allele frequency changes. The method of Mathieson and McVean (2013) describes an approach where selection coefficients and migration are iteratively estimated (p. 975-976 Mathieson and McVean, 2013). Was that used here? Is there evidence of gene flow between locations? This shouldn't confound the ancient/contemporary overlap test, but it could very much confound contemporary selection approaches (e.g. if one population acted as a source and others as sinks for migrants).
- The authors should specify the effective population size estimates in the main text of the paper (not all of Supp. Table S3, but a rough sense of the scale). This is an important bit of information to have, as well as the method used to estimate N_e (e.g. say it's temporal based). I found in the supplementary material they are using the temporal N_e estimates of Epstein et al. (2019) which are $N_e=34$ (as well as their own estimates from spatpg) – this is very low and definitely worth putting in the main text. As Fisher and Ford (1947), Wright (1948), and Mathieson and McVean (2013) discuss, temporal selection estimates are only as good as their N_e estimates, and so this needs to be mentioned in the main text.
- The authors claim that the lack of overlap of selected loci detected in the contemporary and historical selection datasets is evidence of evolutionary novelty. I feel that this claim is too strong in the present paper for one important reason: with a low N_e , drift is very strong and variance in allele frequency changes very high. Sites under historical selection may also be under selection today, but between strong drift (due to low N_e) and sampling noise, may not have been detected in the current study – in other words, the lack of overlap could be due to lack of statistical power rather than evolutionary novelty due to different sites under selection. The best way to resolve this would be to simulate the evolution of these populations (with closely matching demography)

with these small Nes, apply the same methods, and see how well these tests detect convergence when that scenario is being simulated. I fully recognize that writing these simulations in SLiM would be a great deal of work, and this is why I am hesitant to mandate it. But if the authors wish to claim that their results are consistent with novel selection between DFTD and DTF2, these simulations are essential. I believe it would be a worthwhile endeavor and make the paper much stronger.

Minor Comments:

- It would improve readability if details of the contemporary selection data were mentioned in the main part of the paper. Lines 131-134 should include details about how temporal samples were obtained and maybe a quick summary of other details from Table 1. Additionally, the authors need a ballpark number of generations between the years in Table 1, and very importantly, they need to discuss whether overlapping generations means the same individuals were sampled, as this can affect tests of contemporary selection.
- Readers would benefit from a sentence or two in the main text describing and contrasting the methods of Mathieson and McVean (2013) and spatpg.
- I commend the authors for their detailed and well-documented Github repository containing the code used in this study. Nicely done.
- In the future, it would be helpful to provide line numbers for the supplementary text as well.

Decision letter (RSPB-2020-1851.R0)

18-Sep-2020

Dear Dr Hohenlohe:

I am writing to inform you that your manuscript RSPB-2020-1851 entitled "Historical and contemporary signatures of selection in response to transmissible cancer in Tasmanian devils (*Sarcophilus harrisi*)" has, in its current form, been rejected for publication in Proceedings B.

This action has been taken on the advice of referees, who have recommended that substantial revisions are necessary. With this in mind we would be happy to consider a resubmission, provided the comments of the referees are fully addressed. However please note that this is not a provisional acceptance.

Sincerely,
 Professor Gary Carvalho
 mailto: proceedingsb@royalsociety.org

Associate Editor
 Board Member: 1
 Comments to Author:
 Dear Dr. Hohenlohe,

Thank you for submitting this manuscript to the special issue. Your manuscript has been carefully reviewed by three experts. All three reviewers were generally positive about the manuscript, but they also had a number of suggestions for improvement. In particular, all reviewers had concerns about conclusions based on the non-overlap between the SNPs identified as being under selection in the historical and contemporary datasets. Reviewer 3 suggests simulations to provide evidence that the non-overlap is due to selection on different genes as opposed to low power. However, I think that in lieu of doing those simulations, a more careful discussion of the possible causes of non-overlap would be appropriate. In addition, multiple reviewers also noted that the manuscript lacked key details in the main text, so please add these details.

Reviewer(s)' Comments to Author:
 Referee: 1
 Comments to the Author(s)
 My comments to the authors are attached as a pdf file.

Referee: 2
 Comments to the Author(s)

The authors synthesize results from a number of population genomic approaches aimed at detecting selection at different timescales in the Tasmanian Devil, which is facing a presumably novel selective pressure due to transmissible cancer across much of its range. I found the study system and design of the experiment to be very interesting, but I don't think it is suitable for publication in its current form. Particularly, I found the methods and results section to be critically lacking in detail.

Prior to going to the supplemental, I found it hard to understand what was being compared with what and how comparisons were being made. I recognize that space constraints exist, but in this case, I can't even tell from the Methods what criteria are being used to determine candidate SNPs. I don't think a reader should be expected to pull up the Supp. Materials to get this kind of basic information. In terms of raw number of lines, over half of the manuscript is dedicated to the Discussion, so if space is a concern, I think space can be carved out for additional detail by slimming down the Discussion.

The Results are additionally peppered with very vague statements like "The distributions...had an overall low magnitude of dN/dS and a low proportion of sites." Given that these are metrics with meaning for most population genomics people, why not provide the actual numbers? This lack of detail is further reflected in the figures. Figure 3 provides very little information regarding both the absolute and relative magnitudes of the distributions being compared. There is lots of discussion regarding specific genes, pathways, GO categories, etc., but none of this is tied to any of the information in Figure 2.

The main result of the paper seems to be lower than expected overlap between candidate loci identified via historical and contemporary selection analyses. However, given that candidates are defined somewhat arbitrarily, it makes me wonder how robust this result would be to

changes in the criteria used for designating candidates. Also, while much of the results/discussion focus on comparing the current results to prior studies, I often found it difficult at times to clearly identify the novel findings of this study vs what was found in older studies. With regard to contemporary selection, the authors mention in multiple places that this study greatly expands the number of individuals and genetically independent populations from prior studies, but I couldn't get a sense for what samples and populations were unique in the current study, nor could I get a clear sense of how this additional data has allowed for any new insight. Lastly, the mention of genetically independent populations had me hoping for more insight into the extent that the selection response was shared (or not) across populations. After reading the supplemental materials, it seems this may be a limitation of the method, but I was left wondering throughout the manuscript when the authors would address this with the data. Overall, I think there is a lot of good content in the paper, and with additional detail and edits could be made suitable for publication. There are some angles that I think would benefit the would like to see fleshed out, particularly inter-population variability or consistency, but I understand if this is outside of the authors' intended scope and editors' time constraints for the special feature.

Specific comments:

Line 109: Might be nice to give ballpark indication of what is meant by contemporary and historical in this study.

Line 112: RAD-capture panel was introduced in prior paragraph. Seems unnecessary here.

Line 113: shared between what? I couldn't tell if this meant looking for sharing of these signals across populations or if it meant sharing across the timescales.

Line 115: I got hung up here with how the hypotheses are set up and/or contrasted. I think part of what hung me up is that it isn't clear if we are talking about transmissible cancer sensu stricto DFTD or more broadly. It would seem to me that the novelty of the selective force as it pertains to the selection response would come primarily from the specific type of cancer, and there are many ways that even a particular cell lineage can become malignant.

Does non-overlap of selection targets between historical and contemporary selection necessarily imply novelty of transmissible cancer? Cancer related pathways can be so diverse, it wouldn't surprise me if recurrences of transmissible cancer invoked selection responses at different sets of loci. This seems covered by third hypothesis, but doesn't this just mean that non-overlap does not directly connect to novelty vs recurrence?

I guess I don't see the need to tie the interpretation (novelty vs recurrence) into the basic hypothesis being tested: overlap vs non-overlap of genes involved in historic vs recurrent selection. The latter results are interesting in and of themselves, and they certainly have some bearing on novelty, but I find linking them together like this to be somewhat distracting and confusing.

Line 140-148: Would be good to have a bit more of an indication as to the actual criteria that was used to be considered a candidate. It says that the 3 tests were combined, but I can't tell how this was used to determine if gene is a candidate.

Line 142: Why not delta-p?

Line 164: I stumbled on the first sentence, which I think caused me to struggle on understanding the subsequent content. I initially thought the authors were testing whether the "number", as in the count, of selected genes differs from genome-wide background, and I could not understand what genome-wide background could mean in this case. Upon reading a second time, it seems "number" is being used in the sense of 'the collection of' genes under contemporary selection. Consider rewording.

Line 166: Differences between what? After reading full MS, I understand this better now. I think my confusion was persisting from the preceding sentence, where I thought the authors were testing for a difference in the actual number of genes.

Line 183: What percentage of raw SNPs is this?

Line 187: State 1% in methods. I sympathize that these percentile cutoffs are inherently somewhat arbitrary, but how sensitive are results to the cutoff used?

Line 189: How is enrichment being done? Are GO terms from the devil annotation? Are Rapture probes biased toward disease-related genes? Does background gene set take into account these

biases? I ended up finding this info in Supp Mat, but might be nice to have a nod to this in main doc.

Line 200: Again, what was criteria for inclusion? This really ought to be in the main text. Also, why was no multiple test correction done for the historical selection analysis? Is this not typical for this type of analysis? Seems like it would be called for in this case.

Line 211-212: What does overall low magnitude mean? I can't glean from text or figure anything regarding the magnitudes of candidates from genome-wide.

Line 212-214. Provide numbers even if a negative result.

Line 222: How did you correct for multiple testing?

Line 237: Wondering whether sensitivity of historical selection tests is sufficient to claim that a lack of signal justifies the positive assertion that selection has NOT occurred. Seems like more appropriate to say that the contemporary candidates do not show evidence of historical selection.

Line 266: What is meant by genes associated with devil case-control?

Line 270: What populations? Is this a sample size/power issue since this differed between populations?

Line 280: Is PAX3 a devil tumour regression candidate or just a cancer-related gene?

Line 304: Analytical resolution?

Line 311: What is the number? I'm not sure what a "small handful" of bivalve molluscs would be.

Line 349: See comment regarding line 222.

Supplemental:

What is Bonferroni FDR?

I'm confused by the inflation factors. Where are these Z-scores coming from? And shouldn't they be squared when calculating a GIF?

There are a number of typos in the Supplemental text and figure/table captions.

Supp. Table S7: Would be nice to have the p-values from the historical selection analysis for each entry in the table.

Line 459: Reads funny to me.

Figure1B: Would be nice to have an estimate of divergence times at the nodes.

Referee: 3

Comments to the Author(s)

In this study, Stahlke et al. study the rapid evolution of a transmissible cancer in Tasmanian devils. This transmissible cancer has been evolved independently twice (DFTD and DFT2), which the authors use to see whether the same loci were under selection (as a response to the transmissible cancer) in Tasmanian devils. The authors use two approaches: they look for contemporary selection using allele frequency changes over a shorter timescale, as well as looking for signatures of selection using a molecular evolution approach. The authors compare the overlap in the targets of selection, and while they do find some genes under selection across both tests of selection, they find that there was less overlap than expected by chance. Overall, I found this to be a nice study on an important and interesting topic for both evolution and conservation. However, there were a few methodological issues the author should address before I can accept it.

Major Comments:

- One main concern is that their Δ af method is not using Fisher's angular (or arcsine) transformation. The magnitude of allele frequency change is strongly mediated by where the frequency is in the frequency spectrum; allele frequency changes are larger when they are more intermediate. Fisher and Ford (1947) is the original reference, but see Kelly and Hughes (2019, Genetics) for a recent clear application for the arcsine transform. I like the DCMS approach the authors use, and I imagine this sort of averaging across approaches means their findings are robust despite not using the arcsine transform. Still, using raw allele frequency change is not optimal and should be corrected, especially considering they ranked everything to get a pseudo-p-value for their DCMS input. As an aside, it would be interesting to know if their present ranked SNPs are enriched for more intermediate frequencies due to not using the angular transform.

- Lines 141-143: the authors say that they measured Δaf in each population. First, the authors need to specify how these data within each population are combined together to form the pseudo-p-value, in either the main text or Supplementary Materials. Second, this allows for a possibly interesting alternative analysis: why not look for convergence across populations in allele frequency change? Do certain SNPs show convergence in allele frequency change direction, across populations? For example, see the idea of convergence correlations in Buffalo and Coop, 2020, PNAS – this method isn't needed here, but the same idea and signal is worth looking at. Note though that gene flow makes this difficult, which relates to my next point.
- One concern I had is gene flow across the five locations with sampling before/after DTFD. This can also lead to significant allele frequency changes. The method of Mathieson and McVean (2013) describes an approach where selection coefficients and migration are iteratively estimated (p. 975-976 Mathieson and McVean, 2013). Was that used here? Is there evidence of gene flow between locations? This shouldn't confound the ancient/contemporary overlap test, but it could very much confound contemporary selection approaches (e.g. if one population acted as a source and others as sinks for migrants).
- The authors should specify the effective population size estimates in the main text of the paper (not all of Supp.Table S3, but a rough sense of the scale). This is an important bit of information to have, as well as the method used to estimate N_e (e.g. say it's temporal based). I found in the supplementary material they are using the temporal N_e estimates of Epstein et al. (2019) which are $N_e=34$ (as well as their own estimates from spatpg) – this is very low and definitely worth putting in the main text. As Fisher and Ford (1947), Wright (1948), and Mathieson and McVean (2013) discuss, temporal selection estimates are only as good as their N_e estimates, and so this needs to be mentioned in the main text.
- The authors claim that the lack of overlap of selected loci detected in the contemporary and historical selection datasets is evidence of evolutionary novelty. I feel that this claim is too strong in the present paper for one important reason: with a low N_e , drift is very strong and variance in allele frequency changes very high. Sites under historical selection may also be under selection today, but between strong drift (due to low N_e) and sampling noise, may not have been detected in the current study – in other words, the lack of overlap could be due to lack of statistical power rather than evolutionary novelty due to different sites under selection. The best way to resolve this would be to simulate the evolution of these populations (with closely matching demography) with these small N_e s, apply the same methods, and see how well these tests detect convergence when that scenario is being simulated. I fully recognize that writing these simulations in SLiM would be a great deal of work, and this is why I am hesitant to mandate it. But if the authors wish to claim that their results are consistent with novel selection between DTFD and DTF2, these simulations are essential. I believe it would be a worthwhile endeavor and make the paper much stronger.

Minor Comments:

- It would improve readability if details of the contemporary selection data were mentioned in the main part of the paper. Lines 131-134 should include details about how temporal samples were obtained and maybe a quick summary of other details from Table 1. Additionally, the authors need a ballpark number of generations between the years in Table 1, and very importantly, they need to discuss whether overlapping generations means the same individuals were sampled, as this can affect tests of contemporary selection.
- Readers would benefit from a sentence or two in the main text describing and contrasting the methods of Mathieson and McVean (2013) and spatpg.
- I commend the authors for their detailed and well-documented Github repository containing the code used in this study. Nicely done.
- In the future, it would be helpful to provide line numbers for the supplementary text as well.

Author's Response to Decision Letter for (RSPB-2020-1851.R0)

See Appendix A.

RSPB-2021-0577.R0

Review form: Reviewer 1

Recommendation

Accept with minor revision (please list in comments)

Scientific importance: Is the manuscript an original and important contribution to its field?

Excellent

General interest: Is the paper of sufficient general interest?

Good

Quality of the paper: Is the overall quality of the paper suitable?

Good

Is the length of the paper justified?

Yes

Should the paper be seen by a specialist statistical reviewer?

No

Do you have any concerns about statistical analyses in this paper? If so, please specify them explicitly in your report.

No

It is a condition of publication that authors make their supporting data, code and materials available - either as supplementary material or hosted in an external repository. Please rate, if applicable, the supporting data on the following criteria.

Is it accessible?

Yes

Is it clear?

Yes

Is it adequate?

Yes

Do you have any ethical concerns with this paper?

No

Comments to the Author

The manuscript has considerably changed and the authors have incorporated almost all of the suggestions by the reviewers. After re-organization and rewording of many parts the manuscript now ready really well, well done!

The hypotheses put forward in lines 118-129 represents the results of the paper now, however line 125 isn't quite right. In the first two hypotheses (lines 118-124) the basis of response to DFTD may still be polygenic (many underlying loci) but repeatable (the same genes/loci being targeted). For the 3rd hypothesis, if the polygenic response uses the available genetic redundancy different loci will respond to DFTD in contemporary populations. The authors touched upon redundancy in line 357, so it would be good if this part of hypothesis is also reworded.

Line 225-226: it's mentioned that the initial frequencies for candidate SNPs were not skewed toward intermediate frequencies but looking at Fig. S9. it seems that at least for some populations e.g. Fentonbury, Freycinet, and Narawntapu, the candidate SNPs are over-represented in the intermediate frequency bin (i.e. 0.4-0.5). What is the interpretation for this? Are these SNPs under balancing selection?

Fig. 3 Something might have gone wrong probably while converting this figure to pdf because what is described in caption (gray squares, black squares, and red circles) can't be seen in the figure. I only see 16 black dots!

Fig S2. If the folded SFS is shown here, then it should be the frequency of minor allele ranging from 0 to 0.5. Is the SFS shown here not folded?

Fig. S7. The label of x axis should be chromosome.

Review form: Reviewer 3

Recommendation

Accept as is

Scientific importance: Is the manuscript an original and important contribution to its field?

Excellent

General interest: Is the paper of sufficient general interest?

Excellent

Quality of the paper: Is the overall quality of the paper suitable?

Excellent

Is the length of the paper justified?

Yes

Should the paper be seen by a specialist statistical reviewer?

No

Do you have any concerns about statistical analyses in this paper? If so, please specify them explicitly in your report.

No

It is a condition of publication that authors make their supporting data, code and materials available - either as supplementary material or hosted in an external repository. Please rate, if applicable, the supporting data on the following criteria.

Is it accessible?

Yes

Is it clear?

Yes

Is it adequate?

Yes

Do you have any ethical concerns with this paper?

No

Comments to the Author

I commend the authors on their hard work and careful revisions on this nice paper. Good work.

Decision letter (RSPB-2021-0577.R0)

12-Apr-2021

Dear Dr Hohenlohe:

Your manuscript has now been peer reviewed and the reviews have been assessed by an Associate Editor. The reviewers' comments (not including confidential comments to the Editor) and the comments from the Associate Editor are included at the end of this email for your reference. As you will see, the reviewers and the Editors have raised some concerns with your manuscript and we would like to invite you to revise your manuscript to address them. While I agree that the comments remaining are relatively minor, it is important they are addressed, specifically the comments relating to: rewording of the hypothesis, the overrepresentation of intermediate SNP frequencies, and the ambiguously presented figure. Such modification should require minimum time, and we look forward to seeing the revision as soon as you are able.

Research ethics:

Use of animals and field studies:

It is a condition of publication that you make available the data and research materials supporting the results in the article (<https://royalsociety.org/journals/authors/author-guidelines/#data>). Datasets should be deposited in an appropriate publicly available repository and details of the associated accession number, link or DOI to the datasets must be included in the Data Accessibility section of the article (<https://royalsociety.org/journals/ethics-policies/data-sharing-mining/>). Reference(s) to datasets should also be included in the reference list of the article with DOIs (where available).

Please submit a copy of your revised paper within three weeks. If we do not hear from you within this time your manuscript will be rejected. If you are unable to meet this deadline please let us know as soon as possible, as we may be able to grant a short extension.

Best wishes,
Professor Gary Carvalho
<mailto:proceedingsb@royalsociety.org>

Associate Editor Board Member

Comments to Author:

This resubmitted article has been carefully reviewed by two expert reviewers. They both agreed that the manuscript has been greatly improved by the revisions and they had no significant concerns. Please address the revisions suggested by reviewer 2.

Reviewer(s)' Comments to Author:

Referee: 3

Comments to the Author(s).

I commend the authors on their hard work and careful revisions on this nice paper. Good work.

Referee: 1

Comments to the Author(s).

The manuscript has considerably changed and the authors have incorporated almost all of the suggestions by the reviewers. After re-organization and rewording of many parts the manuscript now ready really well, well done!

The hypotheses put forward in lines 118-129 represents the results of the paper now, however line 125 isn't quite right. In the first two hypotheses (lines 118-124) the basis of response to DFTD may still be polygenic (many underlying loci) but repeatable (the same genes/loci being targeted). For the 3rd hypothesis, if the polygenic response uses the available genetic redundancy different loci will respond to DFTD in contemporary populations. The authors touched upon redundancy in line 357, so it would be good if this part of hypothesis is also reworded.

Line 225-226: it's mentioned that the initial frequencies for candidate SNPs were not skewed toward intermediate frequencies but looking at Fig. S9, it seems that at least for some populations e.g. Fentonbury, Freycinet, and Narawntapu, the candidate SNPs are over-represented in the intermediate frequency bin (i.e. 0.4-0.5). What is the interpretation for this? Are these SNPs under balancing selection?

Fig. 3 Something might have gone wrong probably while converting this figure to pdf because what is described in caption (gray squares, black squares, and red circles) can't be seen in the figure. I only see 16 black dots!

Fig S2. If the folded SFS is shown here, then it should be the frequency of minor allele ranging from 0 to 0.5. Is the SFS shown here not folded?

Fig. S7. The label of x axis should be chromosome.

Author's Response to Decision Letter for (RSPB-2021-0577.R0)

See Appendix B.

Decision letter (RSPB-2021-0577.R1)

04-May-2021

Dear Dr Hohenlohe

I am pleased to inform you that your manuscript entitled "Contemporary and historical selection in Tasmanian devils (*Sarcophilus harrisii*) support novel, polygenic response to transmissible cancer" has been accepted for publication in Proceedings B.

Data Accessibility section

Open Access

Paper charges

Sincerely,

Professor Gary Carvalho

Associate Editor:

Comments to Author:

Thank you for thoroughly addressing all of the remaining reviewer comments.

Appendix A

**DEPARTMENT OF
BIOLOGICAL SCIENCES**

College of Science
875 Perimeter Drive MS 3051
Moscow ID 83844-3051

208-885-6280
208-885-7905 [FAX]
uidaho.edu/sci/biology

March 8, 2021

Proceedings B
Submitted online

To the Editor:

We are pleased to submit the revision of our manuscript RSBP-2020-1851, now titled “Comparisons of contemporary and historical selection in Tasmanian devils (*Sarcophilus harrisii*) support novel, polygenic response to Devil Facial Tumor Disease (DFTD).”

Based on previous reviews, we have substantially revised all parts of the manuscript, and it has benefited greatly from the reviewers’ comments. Here we have included the “clean” revised text, a version showing tracked changes, and all supplemental material, along with specific responses to all comments shown in blue below.

Sincerely,

A black rectangular redaction box covering the signature of the author.

in behalf of all authors

Editor

Thank you for submitting this manuscript to the special issue. Your manuscript has been carefully reviewed by three experts. All three reviewers were generally positive about the manuscript, but they also had a number of suggestions for improvement. In particular, all reviewers had concerns about conclusions based on the non-overlap between the SNPs identified as being under selection in the historical and contemporary datasets. Reviewer 3 suggests simulations to provide evidence that the non-overlap is due to selection on different genes as opposed to low power. However, I think that in lieu of doing those simulations, a more careful discussion of the possible causes of non-overlap would be appropriate. In addition, multiple reviewers also noted that the manuscript lacked key details in the main text, so please add these details.

Response: We have substantially revised the manuscript in response to the comments from all reviewers, including the concerns about the degree of overlap between time scales, addressed in detail below. This includes changing the title and significant revisions throughout the text to clarify our results, and what conclusions we can draw. We have addressed the issues of statistical power and genetic drift in better describing our approach to identifying candidate genes. We have also added several revisions to our analyses as suggested by reviewers, including arcsin transform of allele frequencies, multiple testing correction, testing for parallel evolution across contemporary populations, and more described below. Finally, we have revised the Methods section in the main text and the Supplementary Material to provide key details in the main text, while staying with length limits for the journal.

In our responses to reviewers below, we refer to the line numbers of the changes accepted documents.

Reviewer 1

In this manuscript the authors aimed to identify contemporary and historical selection signatures in response to transmissible cancer in Tasmanian Devil. Using a large dataset with samples from different populations and years, the authors have used/merged 3 methods for detection of selected loci. They also identified historical positive selection by comparing Devil genome with other marsupials. I find the topic interesting and important, and specifically the dataset is quite impressive. The manuscript is also well-written and easy to follow. I do have some concerns about the methods used for detection of selection and specifically interpretation of the result of historical selection. In addition, this dataset can be used to investigate (non)parallel response across populations and some population genetics aspects of the observed selection response (comments below).

Major Comments

1. 1982 genes are identified to be under historical positive selection but not all are selected due to DFTD. But with the lack of significant overlap between the contemporary and historical signatures, it isn't feasible to tease apart the genes selected in response to previous DFTD incidences. However, the title of the manuscript suggests that authors have managed to identify historical selection in response to DFTD. It is also mentioned

in the manuscript that due to lack of overlap, the current response to DFTD is different from the historical one, while in fact there is no evidence that DFTD is a recurrent incident in Devils. I suggest authors tone down these conclusions/speculations.

Response: We have made several changes to be more clear about the hypotheses we tested about historical selection and the potential recurrence of transmissible cancer in the devil lineage. We modified the title to “Comparisons of contemporary and historical selection in Tasmanian Devils (*Sarcophilus harrisi*) support novel, polygenic response to Devil Facial Tumour Disease (DFTD).” We have substantially revised several parts of the manuscript to address the reviewer’s concerns. For example, see lines 331-335 of the Discussion and the Conclusion in lines 408-416.

2. Figure 2. in the DeltaAF subplot, why some of the candidate SNPs (filled ones) e.g. the yellow dots for Fentonbury, have very low $-\log_{10}(\text{pvalue})$? Why many SNPs (shown in square) with low p-values are not candidates? In supplementary material, section results, it is mentioned that “Among the three tests for selection signatures: 174 genes were in top 5% of parallel allele shifts following disease (Daf)’, however with all 6 populations shown in 1 plot, it is difficult to see whether the candidate SNPs are localized in similar genomic regions or not. It would be good to see the patterns of allele frequency change for each population separately too. To identify parallel frequency shifts, authors can perform Cochran–Mantel–Haenszel test to find parallel selection response across populations. Fisher’s exact test/chi square test for each population can also identify any population-specific response. Combination of these two types of tests can provide a better understanding of parallel/or different selection across the 6 populations (see comment 4 too). It would be helpful to show the distribution of allele frequency changes, selection coefficient and starting frequencies of the candidate SNPs for each population and for each method including the top 1% DCMS candidates. This would provide further explanation about (non)parallelism selection response across populations.

Response: We have redrawn Figure 2 to make candidate SNPs more clear and revised the caption to address these comments. Specifically, the filled circles across all methods including *delta-af* represent candidate SNPs from the DCMS top 1% in order to reflect the different contributions across methods and populations. Because of this, each subplot has filled points with low $-\log_{10}(\text{p-value})$ because those SNPs were detected as candidates given the full weight of all evidence in the DCMS, even though they were not necessarily candidates within that analysis. We feel this representation illustrates the independence of methods and populations, and the degree of overlap in candidate loci. Additionally, we have added to the supplementary material (Supplementary Figures S3-S4): plots of allele frequency change (*delta-af*) and the Mathieson McVean method (*mm*) for each population separately, with the top 1% of each method shown in opaque points above the dashed threshold line.

To test for repeatability among the responses among populations, we computed a similarity index called the C-score, developed by Yeaman et al 2018

(<https://doi.org/10.1371/journal.pgen.1007717>) which compares the degree of sharing among populations vs unique responses for different populations. We found low but significant parallelism among populations. We now address this issue in the Introduction in lines 58-60, the Methods in lines 163-165, the Results in lines 221-223, and in the Discussion in lines 276-278. The table of overlaps (Supplemental Table S4) also provides some context to the degree of parallelism among populations.

Finally, we have also included visualizations of the folded distribution of starting allele frequencies for each population (Supplemental Figure S2) and a comparison of starting frequencies for candidates and noncandidates from each population (Supplemental Figure S9).

3. In supplementary material, section results, 'Among the three tests for selection signatures: 174 genes were in top 5% of parallel allele shifts following disease (Deltaaf), 68 were in the top 1% of spatpg scores, and 49 were in the top 1% of mm scores (Figure 2).' What is the overlap between these 3 sets of candidate SNPs/genes? How about overlap across populations? Line 187: it is specified that top 1% DCMS score has resulted in 79 annotated/186 genes. What is the overlap between these genes and those that are identified as candidates (top 1-5%) from each of the tests separately?

Response: We added a presentation of the overlap among methods in Supplementary Table S4, in addition to the parallelism analyses described above. Table S4 provides the number of genes associated with the top 1% of SNPs for all pairs of tests and populations.

4. The results of this manuscript suggest that the selection response to DFTD is polygenic, thus redundancy may result in non-parallelism across populations. As mentioned in comment 2, investigating the population-specific selection response can provide more information on this matter. Authors have made an attempt in this line to compare the selection response across populations (Fig. S2 and S3). However, a more quantitative comparison will be more informative. Computing a similarity index such as Jaccard's index or C score (Yeaman et al. 2018, <https://doi.org/10.1371/journal.pgen.1007717>) using the candidate SNPs or genes can provide a quantitative measure of parallelism. The similarity can then be tested against the null hypothesis of random sharing.

Response: We have incorporated the C-score from Yeaman et al (2018) as discussed above. We note, however, that our primary aim in this study was to compare historical versus contemporary selection, not to test for parallelism among contemporary populations. Because of the short timescale of selection by DFTD and our time series samples, and because of the effects of drift in these relatively small devil populations, our power to detect candidate loci within any single population is limited. Therefore, most of our inferences about contemporary selection are based on the composite scores across populations.

5. Line 250-251: (Fig. S2, S3) although I find similarities/correlations among selection signatures of different populations interesting (i.e. similar selection signature among populations), what is the interpretation for the higher similarity between methods rather

than populations? For example, in Figure S3 Fentonbury.mm and Foretier.mm are more similar to each other than Fentonbury.afchange and Foretier.afchange.

Response: We discuss this result briefly on lines 274-278. In general, we found higher correlations among populations with the same method compared to among methods within the same population (Supplementary Table S10). This highlights the fact that different methods pick up different signals in the time series data (within populations), but are still able to identify parallel evolution across populations.

6. The details of analyses for the selection tests for both contemporary and historical data sets are in supplementary material, while these are important steps in data analysis that determine the identified candidates and the drawn conclusions, I suggest transferring them to the very brief Materials and Methods section in the main text. Similarly, parts of Result section in the supplementary material regarding the result of selection analysis and (non)parallelism across populations are quite interesting and important, I suggest to transferring them to the main text.

Response: We have substantially re-structured the Methods and Results to address this comment. While space limitations still require many details to remain in the Supplementary section, more information is now given in the main text.

Minor Comments

7. In supplementary material, section 'Spatial and temporal analysis of contemporary selection', 'we estimated the magnitude and direction of allele frequency change at SNPs with data from at least 10 individuals at both time points'. The estimated disease fronts are indicated in Figure 1, but it would be helpful if the date used to classify before and after datasets (i.e. the year of occurrence of DFTD) for each population is mentioned. Also, based on this description it seems that only 2 time points were used for computing allele frequency change. Would using different time points result in higher magnitude of frequency change? Providing more details for this part is appreciated.

Response: We have re-written these sections in both the Methods and Supplementary material to be more clear. The year of first detection and before/after sample sizes are shown in Table 1. The Mathieson-McVean and spatpg analyses use multiple time points, providing the contrast with the use of 2 time points for allele frequency change, as suggested by the reviewer, so that part of the difference among methods reflects this issue of binning time points.

8. In supplementary material, section 'Spatial and temporal analysis of contemporary selection', 'The SNPs were ranked by the magnitude of change, and the fractional rank was used as a pseudo- p-value for the composite statistic (described below).' What is the rationale for using this method and dealing with pseudo p-values instead of using tests such as Fisher's exact or Chi square test to get p-values?

Response: We acknowledge there are a number of ways to identify SNPs with significant allele frequency changes following DFTD. We chose to use the fractional rank approach to avoid the issue of multiple testing across loci, time-points, and populations. Additionally, the incorporation of Fisher's arc-sin transformation suggested by Reviewer 3, further mitigates the effect of allele frequency spectra, providing a more appropriate distribution of p-values.

9. Regarding the inflation rates for mm and spatpg analysis, it would be good if authors show the p-values with and without inflation rate corrections to see how the distribution of p-values differ before and after correction.

Response: We have now included these in Figures S3 and S5.

10. These populations have very low N_e and probably high drift, what are the chances that some of the allele frequency changes are due to drift?

Response: We chose to use methods that account for effective population sizes (*mm* and *spatpg*), as well as the composite statistic (DCMS) which used multiple lines of evidence because of the low effective population sizes and relative strength of drift compared to selection. Together, these methods mitigate the effect of drift and conservatively identify SNPs very likely to be associated with selection due to DFTD. We clarify this issue in line 144.

11. In supplementary material, section 'Functional Enrichment of Genes Under Selection' paragraph: 'We considered the overrepresentation result statistically significant with p-values < 0.001'. I didn't find any mention of multiple testing correction in methods, have this been done? And why not do permutation test (similar to what is done for comparison of contemporary and historical candidates) instead of an ad hoc p-value?

Response: Previously we had not corrected for multiple testing in the permutation test for gene set enrichment analysis with MSigDB. We appreciate the reviewer's suggestion to perform multiple testing, and did so using the method of Benjamini-Hochberg as suggested by the reviewer in a later comment (https://github.com/Astahlke/contemporary_historical_sel_devils/blob/master/compare/test_for_ongoing_selection.R). After correcting for multiple testing, we did not find shared enrichment for gene sets between contemporary and historical selection. We added language in lines 155-157 of the Supplemental Methods and in lines 261 of the Main Text Results. We found this to be a conventional and straight-forward method to examine the evidence in a long list of gene set enrichment tests, rather than a permutation test.

12. Figure 3: Line 200 specified finding 1,982 genes with a signal of historical positive selection, and the caption of figure 3 suggests that what is labeled as 'the genome-wide background' is the 1982 genes with historical positive selection. Is there a specific reason for not using the label: 'historical selection'? What is the number of genes shared between contemporary and historical selection? It is 21 in line 208 but 19 in other

sections. Are having such high dNdS values normal? Were synonymous sites kept in the alignment before computing dNdS?

Response: We have revised Figure 3, and it now presents all genes detected under historical positive selection both in distribution of dN/dS metrics and across the genome. We have reworded the caption to reflect this and the reviewer's comments. The tests for historical selection specifically identify positive selection (e.g. $dN/dS > 1$), so using the term positive selection is accurate. We have clarified the language about the number of recurrent selection candidates. We compare our dN/dS results to other published work (lines 314-315). All coding sequence (synonymous and non-synonymous) for orthologs were kept in the alignments to calculate dN/dS statistics.

13. Line 271-273. There is very little overlap between this study and previous ones, considering that the current study has a more comprehensive dataset and has targeted more loci in RAPTURE. Could the extent of overlap depend on the method used for identification of selection targets?

Response: Yes, as described above we expect these tests for contemporary selection to be noisy due to small effective population sizes and relatively few generations of evolution. Different methods for detecting loci under selection also commonly produce quite different results because of their different assumptions. Additionally, here we compare results across studies that focused on different signals: identification of candidate loci under selection, loci identified in GWAS of disease-related traits, and loci associated with tumor regression. For all of these reasons, our approach within this study and in comparing to other studies is to focus on loci that are identified by multiple independent lines of evidence as the strongest candidates for functional significance.

14. Line 344. 22 gene sets are not significant. After multiple testing correction, only one enriched gene set is left. I noticed that bonferroni correction was used which is very conservative. Have authors tried using Benjamini-Hochberg method for correction?

Response: We appreciate the reviewer's suggestion to use the Benjamini-Hochberg method for multiple testing correction. However, after we applied correction for multiple testing across the test for historical selection as suggested by Reviewer 2 and re-ran the test for shared gene sets, we did not find any shared gene sets enriched.

15. Some causal loci might go undetected due to the lack of availability of whole genome sequences. It would be good if authors mention this as part of caveats/issues in the discussion.

Response: We address this issue in lines 323-325.

16. Line 25-29: Only 1 shared gene set is significant, I don't think 22 should be mentioned when not significant.

Response: Corrected. See response to comment 14.

17. Longer genes have more SNPs. Does SNP2GO correct for gene length?

Response: Yes, SNP2GO accounts for this bias by comparing the observed GO associations for candidate SNPs within a region to a generated null distribution of expected GO term associations. To generate this null distribution, SNP2GO performs a default of 1000 runs of hypergeometric samples from the provided background list of SNPs (Szkiba et al 2014).

18. What does MORF stand for?

Response: MORF is an unpublished compendium of cancer gene sets by the Broad Institute (Subramanian et al 2005). It is not actually defined what MORF stands for.

Reviewer 2

The authors synthesize results from a number of population genomic approaches aimed at detecting selection at different timescales in the Tasmanian Devil, which is facing a presumably novel selective pressure due to transmissible cancer across much of its range. I found the study system and design of the experiment to be very interesting, but I don't think it is suitable for publication in its current form. Particularly, I found the methods and results section to be critically lacking in detail.

Major comments:

Prior to going to the supplemental, I found it hard to understand what was being compared with what and how comparisons were being made. I recognize that space constraints exist, but in this case, I can't even tell from the Methods what criteria are being used to determine candidate SNPs. I don't think a reader should be expected to pull up the Supp. Materials to get this kind of basic information. In terms of raw number of lines, over half of the manuscript is dedicated to the Discussion, so if space is a concern, I think space can be carved out for additional detail by slimming down the Discussion.

Response: We have substantially re-organized the Methods and Supplemental material to bring more useful information into the main text. We have also trimmed down the Discussion as suggested.

The Results are additionally peppered with very vague statements like "The distributions...had an overall low magnitude of dN/dS and a low proportion of sites." Given that these are metrics with meaning for most population genomics people, why not provide the actual numbers?

Response: We have added quantitative details here and elsewhere in the Results (e.g. see lines 243-244 and lines 255-256)

This lack of detail is further reflected in the figures. Figure 3 provides very little information regarding both the absolute and relative magnitudes of the distributions being compared. There is lots of discussion regarding specific genes, pathways, GO categories, etc., but none of this is tied to any of the information in Figure 2.

Response: We have revised these figures. Specifically, both Figure 2 and Figure 3 show absolute and relative distributions of the test statistics for the two time scales. The discussion of candidate genes and pathways does not apply directly to these Figures, but is presented elsewhere in the Results, Discussion, and supplementary figures and tables.

The main result of the paper seems to be lower than expected overlap between candidate loci identified via historical and contemporary selection analyses. However, given that candidates are defined somewhat arbitrarily, it makes me wonder how robust this result would be to changes in the criteria used for designating candidates.

Response: At both timescales, we apply rigorous statistical tests to identify candidate loci. Changing the tests used or the thresholds applied for significance would change the sets of candidates, but as in any genomic study this is a balance between false positive and false negative results. Our approach here was to integrate across multiple lines of evidence. However, as the reviewer notes we found little overlap in candidate loci between the two time scales, and we believe this result is fairly robust to the specific methods or significance thresholds we applied.

Also, while much of the results/discussion focus on comparing the current results to prior studies, I often found it difficult at times to clearly identify the novel findings of this study vs what was found in older studies. With regard to contemporary selection, the authors mention in multiple places that this study greatly expands the number of individuals and genetically independent populations from prior studies, but I couldn't get a sense for what samples and populations were unique in the current study, nor could I get a clear sense of how this additional data has allowed for any new insight.

Response: Yes, but more comprehensive sample size, time series, populations is better for gene lists and conversation monitoring - but what's truly novel here is the comparison with historical selection.

Lastly, the mention of genetically independent populations had me hoping for more insight into the extent that the selection response was shared (or not) across populations. After reading the supplemental materials, it seems this may be a limitation of the method, but I was left wondering throughout the manuscript when the authors would address this with the data.

Response: Although the focus of this manuscript is the comparison of contemporary and historical signatures of selection, we have incorporated some analysis (lines 163-165, 221-223) and discussion of population-specific responses (lines 276-278). Also, please see responses to Reviewer 1.

Overall, I think there is a lot of good content in the paper, and with additional detail and edits could be made suitable for publication. There are some angles that I think would benefit the would like to see fleshed out, particularly inter-population variability or consistency, but I understand if this is outside of the authors' intended scope and editors' time constraints for the special feature.

Specific comments:

1. Line 109: Might be nice to give ballpark indication of what is meant by contemporary and historical in this study.

Response: We added "6-8 generations" and "65-85 million years" in lines 109-110 to provide a specific ballpark of contemporary and historical time scales.

2. Line 112: RAD-capture panel was introduced in prior paragraph. Seems unnecessary here.

Response: Removed this second redundant description of the RADcapture panel.

3. Line 113: shared between what? I couldn't tell if this meant looking for sharing of these signals across populations or if it meant sharing across the timescales.

Response: We clarified the language of our hypotheses in the concluding paragraph of our introduction (lines 118-129). In this case, we rephrase to refer to "a conserved response among populations and an overrepresentation of the same genes or pathways under both contemporary historical selection." We argue that both lines of comparison, at population-scale and time-scale, can inform our inference of novelty.

4. Line 115: I got hung up here with how the hypotheses are set up and/or contrasted. I think part of what hung me up is that it isn't clear if we are talking about transmissible cancer sensu stricto DFTD or more broadly. It would seem to me that the novelty of the selective force as it pertains to the selection response would come primarily from the specific type of cancer, and there are many ways that even a particular cell lineage can become malignant. Does non-overlap of selection targets between historical and contemporary selection necessarily imply novelty of transmissible cancer? Cancer related pathways can be so diverse, it wouldn't surprise me if recurrences of transmissible cancer invoked selection responses at different sets of loci. This seems covered by third hypothesis, but doesn't this just mean that non-overlap does not directly connect to novelty vs recurrence? I guess I don't see the need to tie the interpretation (novelty vs recurrence) into the basic hypothesis being tested: overlap vs non-overlap of genes involved in historic vs recurrent selection. The latter results are interesting in and of themselves, and they certainly have some bearing on novelty, but I find linking them together like this to be somewhat distracting and confusing.

Response: We agree that these hypotheses are not necessarily mutually exclusive and our interpretations cannot be conclusive. We have worked to clarify the hypotheses outlined in this paragraph (lines 118-129) and our interpretation of the results in the Discussion (lines 331-338, 408-414).

5. Line 140-148: Would be good to have a bit more of an indication as to the actual criteria that was used to be considered a candidate. It says that the 3 tests were combined, but I can't tell how this was used to determine if gene is a candidate.

Response: We have clarified how the three analyses were combined for the final list of contemporary candidates in lines 159-160.

6. Line 142: Why not delta-p?

Response: We use delta-af (allele frequency) to avoid confusion with p-values.

7. Line 164: I stumbled on the first sentence, which I think caused me to struggle on understanding the subsequent content. I initially thought the authors were testing whether the "number", as in the count, of selected genes differs from genome-wide background, and I could not understand what genome-wide background could mean in this case. Upon reading a second time, it seems "number" is being used in the sense of 'the collection of' genes under contemporary selection. Consider rewording.

Response: Done (lines 190-193). We intended to refer to the collection of genes.

8. Line 166: Differences between what? After reading full MS, I understand this better now. I think my confusion was persisting from the preceding sentence, where I thought the authors were testing for a difference in the actual number of genes.

Response: In addition to re-wording the previous sentence, we clarified this sentence as well to be more specific in referring to the differences in magnitude of dN/dS for historical and recurrent candidates (lines 193-197).

9. Line 183: What percentage of raw SNPs is this?

Response: There is not an easy response to this request because the starting point for the number of raw SNPs could be defined pre-filtering, within or among cohorts-population combinations, etc.

10. Line 187: State 1% in methods. I sympathize that these percentile cutoffs are inherently somewhat arbitrary, but how sensitive are results to the cutoff used?

Response: We added the 1% cut-off, DCMS score = 1.167, to the results in line 223. Yes, the number of candidates detected depends on the designation of this cut-off, but we believe the

top-level results comparing populations and time scales to be robust (See response to main comment above).

11. Line 189: How is enrichment being done? Are GO terms from the devil annotation? Are Rapture probes biased toward disease-related genes? Does background gene set take into account these biases? I ended up finding this info in Supp Mat, but might be nice to have a nod to this in main doc.

Response: Due to space limitations, we describe this in full detail in the supplemental materials, but included the clause “using the subset of genes tested for each test as the respective background set” in lines 200-201 to acknowledge the bias in our sampling.

12. Line 200: Again, what was criteria for inclusion? This really ought to be in the main text. Also, why was no multiple test correction done for the historical selection analysis? Is this not typical for this type of analysis? Seems like it would be called for in this case.

Response: We added text to describe the criteria for inclusion in lines 239-242. Additionally, we added correction for multiple testing here, applying the Benjamini-Hochberg correction (as in the gene set analysis).

13. Line 211-212: What does overall low magnitude mean? I can't glean from text or figure anything regarding the magnitudes of candidates from genome-wide.

Response: We added text to describe the magnitudes of dN/dS estimates for the genome-wide historical candidates (lines 242-244) and candidates for recurrent selection (lines 255-256).

14. Line 212-214. Provide numbers even if a negative result.

Response: Done. We provided the p-values for each test of equality in lines 257-258.

15. Line 222: How did you correct for multiple testing?

Response: We updated our analysis to include correcting for multiple testing with the method of Benjamini-Hochberg, the main text to note this briefly (line 261), and the Supplemental to include details (lines 155-157).

16. Line 237: Wondering whether sensitivity of historical selection tests is sufficient to claim that a lack of signal justifies the positive assertion that selection has NOT occurred. Seems like more appropriate to say that the contemporary candidates do not show evidence of historical selection.

Response: We have modified the text of this conclusion in lines 331-336 and lines 354-363 to be more specific and nuanced as the reviewer (and Reviewer 1) suggested.

17. Line 266: What is meant by genes associated with devil case-control?

Response: We re-worded to clarify, “genes associated with devil infection status (case-control)” at line 289.

18. Line 270: What populations? Is this a sample size/power issue since this differed between populations?

Response: We added the list of populations included in both studies (293-294). We believe that our study with more individuals and populations is more robust, and that both the validation of five candidate genes and elimination of two candidate genes from that study are improvements to the list of candidate genes involved with the contemporary response.

19. Line 280: Is PAX3 a devil tumour regression candidate or just a cancer-related gene?

Response: We added text in this paragraph (lines 300-301) to clarify that the specific genes discussed, including PAX3, are candidates for regression in devils specifically. PAX3 is notable because it has an obvious angiogenesis annotation.

20. Line 304: Analytical resolution?

Response: We re-phrased this sentence (lines 321-325), “ Our ability to detect historical selection due to transmissible cancer in devils could be improved by genome assembly efforts among more closely-related Dasyuridae, as well as complementary annotation.”

21. Line 311: What is the number? I’m not sure what a “small handful” of bivalve molluscs would be.

Response: We re-worded this sentence to focus on the paucity of transmissible cancers in nature (lines 328-330). So far, the transmissible cancer in bivalves has been described in four species, but it is thought that more species may be vulnerable (Yonemitsu et al 2019).

22. Line 349: See comment regarding line 222.

Response: We clarified the procedure to correct for multiple testing with the method of Benjamini-Hochberg to the Gene Set Enrichment Analysis in the Supplemental Methods (lines 155-157) and main text Results (lines 261-265).

Supplemental:

23. What is Bonferroni FDR?

Response: This is referring to the figure caption for the table of MSigDB gene sets significantly enriched by contemporary and historical candidates; however this table no longer exists. After correcting the PAML results for multiple testing with the Benjamini-Hochberg method, re-running

the gene set enrichment test, and correcting again for multiple testing with the Benjamini-Hochberg method, we did not recover any shared gene sets.

24. I'm confused by the inflation factors. Where are these Z-scores coming from? And shouldn't they be squared when calculating a GIF?

Response: The z-scores are derived from the p-values of each test, using the lower tail of the chi square quantile distribution. By using the quantile chi square distribution in this way, the z-scores are returned "squared", equivalent to squaring the p-values first. The code to perform this calculation is provided at https://github.com/Astahlke/contemporary_historical_sel_devils/blob/master/contemporary/script/genomic_inflation_correction.r for review.

25. There are a number of typos in the Supplemental text and figure/table captions.

Response: We have reviewed the Supplemental more carefully to remove typos in the figure and table captions.

26. Supp. Table S7: Would be nice to have the p-values from the historical selection analysis for each entry in the table.

Response: Done.

27. Line 459: Reads funny to me.

Response: We have re-written the figure legend of Figure 3 to improve clarity.

28. Figure1B: Would be nice to have an estimate of divergence times at the nodes.

Response: Done

Reviewer 3

In this study, Stahlke et al. study the rapid evolution of a transmissible cancer in Tasmanian devils. This transmissible cancer has been evolved independently twice (DFTD and DFT2), which the authors use to see whether the same loci were under selection (as a response to the transmissible cancer) in Tasmanian devils. The authors use two approaches: they look for contemporary selection using allele frequency changes over a shorter timescale, as well as looking for signatures of selection using a molecular evolution approach. The authors compare the overlap in the targets of selection, and while they do find some genes under selection across both tests of selection, they find that there was less overlap than expected by chance. Overall, I found this to be a nice study on an important and interesting topic for both evolution and conservation. However, there were a few methodological issues the author should address before I can accept it.

Major comments:

1. One main concern is that their Δf method is not using Fisher's angular (or arcsine) transformation. The magnitude of allele frequency change is strongly mediated by where the frequency is in the frequency spectrum; allele frequency changes are larger when they are more intermediate. Fisher and Ford (1947) is the original reference, but see Kelly and Hughes (2019, Genetics) for a recent clear application for the arcsine transform. I like the DCMS approach the authors use, and I imagine this sort of averaging across approaches means their findings are robust despite not using the arcsine transform. Still, using raw allele frequency change is not optimal and should be corrected, especially considering they ranked everything to get a pseudo-p-value for their DCMS input. As an aside, it would be interesting to know if their present ranked SNPs are enriched for more intermediate frequencies due to not using the angular transform.

Response: We appreciate the referee's valuable, instructional concern. We have incorporated the arcsine transform in our calculation of allele frequency change calculation. This can be seen on the code's github repository:

https://github.com/Astahlke/contemporary_historical_sel_devils/blob/master/contemporary/script/afchange_2020-10-21.sh and is reflected in the Methods section of the Supplemental Information in lines 51-53. However, our DCMS approach was indeed robust. Very few of the final list of candidate genes changed between analyses, and no qualitative result changed, i.e. we found little evidence of historical selection among contemporary candidates.

2. Lines 141-143: the authors say that they measured Δf in each population. First, the authors need to specify how these data within each population are combined together to form the pseudo-p-value, in either the main text or Supplementary Materials. Second, this allows for a possibly interesting alternative analysis: why not look for convergence across populations in allele frequency change? Do certain SNPs show convergence in allele frequency change direction, across populations? For example, see the idea of convergence correlations in Buffalo and Coop, 2020, PNAS — this method isn't needed here, but the same idea and signal is worth looking at. Note though that gene flow makes this difficult, which relates to my next point.

Response: We added additional text to describe how the data for individuals were combined into before and after cohorts in the main text (lines 152-155), columns to Table 1 for the number of samples studied before/after DFTD, the arc-sin transformation of allele frequency change in the supplemental (lines 51-53), and how the fractional rank of SNPs within each population were incorporated into the DCMS final candidate selection in the main text (lines 159-161). For information regarding how the genotypes of individual samples were combined into cohorts (e.g. before and after), we point the reviewer to lines 42-49 of the Supplemental.

We have also incorporated a preliminary analysis of convergence (or the lack thereof) among populations with the C-score (Yeaman et al). We added the motivation in lines 58-60, methods in lines 163-166, and reported the results in lines 220-223. New code for this analysis can be found at

https://github.com/Astahlke/contemporary_historical_sel_devils/blob/master/contemporary/script/dgconstraint.R. We acknowledge this is a preliminary analysis of repeatability and incorporate this in the context of the focus of our work in comparing the signatures of selection across contemporary and historical scales (lines 276-280).

3. One concern I had is gene flow across the five locations with sampling before/after DTFD. This can also lead to significant allele frequency changes. The method of Mathieson and McVean (2013) describes an approach where selection coefficients and migration are iteratively estimated (p. 975-976 Mathieson and McVean, 2013). Was that used here? Is there evidence of gene flow between locations? This shouldn't confound the ancient/contemporary overlap test, but it could very much confound contemporary selection approaches (e.g. if one population acted as a source and others as sinks for migrants).

Response:

At the time-scale of this analysis (6-8 generations), it is unlikely that migrants between populations have driven the selective response to DTFD. We know from an analysis of population structure of many of the same animals that the overall population structure among the same study sites was unchanged; i.e. $K=6$ before and after (Fraik et al 2020). This is further supported by a contraction of home ranges following population declines, attributed to decreases in density of individuals yielding increasing local resources (Comte 2020). Because of this, we chose to implement the Mathieson McVean method analyzing each population independently rather than all populations at once and cannot directly estimate migration among populations with this method.

4. The authors should specify the effective population size estimates in the main text of the paper (not all of Supp.Table S3, but a rough sense of the scale). This is an important bit of information to have, as well as the method used to estimate N_e (e.g. say it's temporal based). I found in the supplementary material they are using the temporal N_e estimates of Epstein et al. (2019) which are $N_e=34$ (as well as their own estimates from spatpg) — this is very low and definitely worth putting in the main text. As Fisher and Ford (1947), Wright (1948), and Mathieson and McVean (2013) discuss, temporal selection estimates are only as good as their N_e estimates, and so this needs to be mentioned in the main text.

Response: Yes, we appreciate the reviewer's comment regarding the role of effective population size and agree that this is an important consideration. We have added the estimates and methods of effective population size to the main text (lines 155-157), "... estimates of effective population size, which ranged from 26-37 according to the two-sample temporal method (51) of NeEstimator v2.01 (52) ..."

5. The authors claim that the lack of overlap of selected loci detected in the contemporary and historical selection datasets is evidence of evolutionary novelty. I feel that this claim is too strong in the present paper for one important reason: with a low N_e , drift is very strong and variance in allele frequency changes very high. Sites under historical selection may also be under selection today, but between strong drift (due to low N_e) and sampling noise, may not have been detected in the current study — in other words, the lack of overlap could be due to lack of statistical power rather than evolutionary novelty due to different sites under selection. The best way to resolve this would be to simulate the evolution of these populations (with closely matching demography) with these small N_e s, apply the same methods, and see how well these tests detect convergence when that scenario is being simulated. I fully recognize that writing these simulations in SLiM would be a great deal of work, and this is why I am hesitant to mandate it. But if the authors wish to claim that their results are consistent with novel selection between DFTD and DTF2, these simulations are essential. I believe it would be a worthwhile endeavor and make the paper much stronger.

Response: We acknowledge that contemporary tests have low power and are subject to noise due to drift, and have worked here to mitigate these effects by 1) maximizing sample size and populations, and 2) choosing methods to account for effective population size (*mm* and *spatpg*) and 3) combining these methods with the composite statistic (DCMS) rather than relying on a single statistical approach.

Our current study is motivated in part by the idea that devils may be vulnerable to transmissible cancers, evidenced by a second independent transmissible cancer, but the populations we studied were not subject to DFT2. Currently DFT2 is restricted to the far southeast corner of Tasmania and has not been found in many devils (Kwon et al 2018). Unfortunately, we could not and did not compare selection between DFTD and DFT2. An important and interesting future direction of this work should examine evolution of devils in response to DFT2 and could compare loci under selection by the two independent transmissible cancers. Simulations modeling the transmission of alleles, individuals, and tumors will be a critical, challenging component of that study. We added a nod to this line of research in lines 397-400.

Minor Comments

6. It would improve readability if details of the contemporary selection data were mentioned in the main part of the paper. Lines 131-134 should include details about how temporal samples were obtained and maybe a quick summary of other details from Table 1. Additionally, the authors need a ballpark number of generations between the years in Table 1, and very importantly, they need to discuss whether overlapping generations means the same individuals were sampled, as this can affect tests of contemporary selection.

Response: We acknowledge that we omitted critical information regarding the origins of samples. Due to limitations in space, and to limit redundancy in the literature, we added the citations to articles (Hamede et al 2014; Fraik et al 2020), which have described well the

extensive field work and sample acquisition performed by our colleagues (lines 235-237). Although generations do overlap in devils, the individual sampled in our study are uniquely identified with a microchip, so we can be very confident that recaptured individuals do not contribute redundantly to our analysis of contemporary evolution. We added columns to Table 1 to document the numbers of samples from each population before and after disease, the parenthetical "(6-8 generations)" in line 109-110, and the word "unique" to line 134.

7. Readers would benefit from a sentence or two in the main text describing and contrasting the methods of Mathieson and McVean (2013) and spatpg.

Response: After addressing the above comments, we did not have space to contrast these methods in the main text beyond their model constraints.

8. I commend the authors for their detailed and well-documented Github repository containing the code used in this study. Nicely done.

Response: Thank you, Reviewer!

9. In the future, it would be helpful to provide line numbers for the supplementary text as well.

Response: Done.

References

Yonemitsu, M.A., Giersch, R.M., Polo-Prieto, M., Hammel, M., Simon, A., Cremonte, F., Avilés, F.T., Merino-Véliz, N., Burioli, E.A., Muttray, A.F. and Sherry, J., 2019. A single clonal lineage of transmissible cancer identified in two marine mussel species in South America and Europe. *Elife*, 8, p.e47788.

Kwon, Y.M., Stammnitz, M.R., Wang, J., Swift, K., Knowles, G.W., Pye, R.J., Kreiss, A., Peck, S., Fox, S., Pemberton, D. and Jones, M.E., 2018. Tasman-PCR: A genetic diagnostic assay for Tasmanian devil facial tumour diseases. *Royal Society open science*, 5(10), p.180870.

Appendix B

**DEPARTMENT OF
BIOLOGICAL SCIENCES**

College of Science
875 Perimeter Drive MS 3051
Moscow ID 83844-3051

208-885-6280
208-885-7905 [FAX]
uidaho.edu/sci/biology

April 26, 2021

Gary Carvalho
Proceedings B
Submitted online

Dear Dr. Carvalho:

We are pleased to submit the final revision of our manuscript RSBP-2021-0577, now titled “Contemporary and historical selection in Tasmanian devils (*Sarcophilus harrisii*) support novel, polygenic response to transmissible cancer.”

We greatly appreciate the detailed feedback and positive comments from both reviewers and the editors. We have addressed the reviewer’s comments specifically (in blue) in the attached response.

Sincerely,

on behalf of all authors

Response to Reviewers' Comments

Associate Editor Board Member

Comments to Author:

This resubmitted article has been carefully reviewed by two expert reviewers. They both agreed that the manuscript has been greatly improved by the revisions and they had no significant concerns. Please address the revisions suggested by reviewer 2.

Reviewer(s)' Comments to Author:

Referee: 3

Comments to the Author(s).

I commend the authors on their hard work and careful revisions on this nice paper. Good work.

Response: We appreciate the reviewer's efforts in helping us improve the manuscript.

Referee: 1

Comments to the Author(s).

The manuscript has considerably changed and the authors have incorporated almost all of the suggestions by the reviewers. After re-organization and rewording of many parts the manuscript now reads really well, well done!

The hypotheses put forward in lines 118-129 represent the results of the paper now, however line 125 isn't quite right. In the first two hypotheses (lines 118-124) the basis of response to DFTD may still be polygenic (many underlying loci) but repeatable (the same genes/loci being targeted). For the 3rd hypothesis, if the polygenic response uses the available genetic redundancy different loci will respond to DFTD in contemporary populations. The authors touched upon redundancy in line 357, so it would be good if this part of hypothesis is also reworded.

Response: We have re-worded the third hypothesis in lines 123-126: "However, if there are multiple, redundant, genetic pathways that could be involved in a response to recurrent transmissible cancers, we may expect a novel response across contemporary populations and little overlap between contemporary and historical timescales."

Line 225-226: it's mentioned that the initial frequencies for candidate SNPs were not skewed toward intermediate frequencies but looking at Fig. S9. it seems that at least for some populations e.g. Fentonbury, Freycinet, and Narawntapu, the candidate SNPs are over-represented in the intermediate frequency bin (i.e. 0.4-0.5). What is the interpretation for this? Are these SNPs under balancing selection?

Response: We have added text to lines 225-228: “The initial frequencies for candidate SNPs were not consistently skewed toward intermediate frequencies across all populations (Supplemental Figure S9). The skew we observed in a few populations (e.g., Fentonbury, Narawntapu) may reflect differences in the power to detect selection with low versus intermediate minor allele frequencies within those populations, or possibly other factors such as balancing selection (52).” We acknowledge that an overrepresentation of intermediate allele frequencies could indicate balancing selection, but without further information or analyses, we sought to avoid speculation and focus our discussion on the broader comparison between contemporary and historical signatures.

Fig. 3 Something might have gone wrong probably while converting this figure to pdf because what is described in caption (gray squares, black squares, and red circles) can't be seen in the figure. I only see 16 black dots!

Response: The figure has been corrected.

Fig S2. If the folded SFS is shown here, then it should be the frequency of minor allele ranging from 0 to 0.5. Is the SFS shown here not folded?

Response: The reviewer is correct. The caption now reads “Figure S2. The allele frequency spectra for the minor allele (as identified in the global dataset) for each population before DFTD became prevalent.”

Fig. S7. The label of x axis should be chromosome.

Response: Figures S7 and S8 have been corrected.